

# A meteorological overview of the ORACLES (ObseRvations of Aerosols above CLouds and their intEractionS) campaign over the southeast Atlantic during 2016-2018

Ju-Mee Ryoo[1,2], Leonhard Pfister[1], Rei Ueyama[1], Paquita Zuidema[3], Robert Wood[4], Ian Chang[5],

Jens Redemann[5]

[1] Earth Science Division, NASA Ames Research Center, Moffett Field, CA, USA

[2] Science and Technology Corporation, Moffett Field, CA, USA

[3] Department of Atmospheric Sciences, Rosenstiel School of Marine and Atmospheric Science, University of Miami, Miami, FL, USA

[4] Department of Atmospheric Sciences, University of Washington, Seattle, WA, USA

[5] University of Oklahoma, School of Meteorology, Norman, OK, USA

*Correspondence to:* Ju-Mee Ryoo (ju-mee.ryoo@nasa.gov)

**Abstract.** In 2016–2018, the ObsErvation of Aerosols above CLouds and their intEractionS (ORACLES) project undertook three major field campaigns in the Southeast (SE) Atlantic Ocean using research aircraft to better understand the impact of biomass burning (BB) aerosol transport to the SE Atlantic Ocean on climate. In particular, ORACLES was designed to

investigate how BB aerosols interact with oceanic stratocumulus clouds, and how that interaction affects the radiation budget. Here, a meteorological analysis has been performed to support the interpretation of the airborne measurements for aerosol transport and its interaction with clouds during the deployments of September 2016, August 2017, and October 2018.

The southern African easterly jet (AEJ-S), represented by the zonal wind at 600–700 hPa < -6 m s$^{-1}$ with entrance over land and exit over the ocean around 5–15° S, is a dominant feature of the mid-level circulation over western Africa during

austral winter and spring (August, September, and October). AEJ-S develops at lower altitudes (~3 km, 700 hPa) in the north



(5–10° S) in August, while it develops at around 4 km (~600 hPa) in the south (5–15° S) in September and October, largely driven by the strong sensible heating over the African plateau. AEJ-S advects air plumes that are both aerosol-laden and moist, implying a clear potential impact by both on SE Atlantic stratocumulus. Benguela low-level jet (LLJ) also develops off the Namibian coast in the SE Atlantic for all three months. Low-level cloud fraction (low-CF) is positively associated with low-level tropospheric stability (LTS) and negatively associated with boundary layer height (BLH). This relationship is especially strong in September and weaker in August and October. Correlation analysis indicates that simple relationships among low-CF, LTS, and BLH break up when rapidly varying large-scale flow and mid-latitude frontal systems intrude.

There are some notable meteorological anomalous characteristics for the three deployment months compared to the climatology: 1) During August 2017, the AEJ-S is slightly weaker than the climatological mean with an anomalous upper-level jet aloft (~6 km) around 10° S. The AEJ-S strength and moisture transport offshore increase at the end of August 2017. August 2017 is also drier over the SE Atlantic than climatology, with a stronger (by ~2 m s$^{-1}$) LLJ. The large-scale anticyclone associated with St. Helena high in the SE Atlantic is stronger and closer to the coast than the climatological mean. 2) During September 2016, the AEJ-S intensity is slightly weaker than the climatological mean and the mid-level RH is slightly higher than the climatological mean, although differences are small. The LLJ is stronger (~1 m s$^{-1}$) than the climatological mean. The large-scale anticyclone associated with St. Helena high in the SE Atlantic is stronger than the climatological mean. 3) During October 2018, the AEJ-S is slightly weaker than the climatological mean and slightly wetter compared to the climatological mean. LLJ is also weaker compared to the climatological mean. The large-scale anticyclone associated with St. Helena high in the SE Atlantic is slightly weaker and further southeast than the climatological mean. Precipitation regions migrate southward from August through October due to seasonal change. During all the deployment years, the sea surface temperatures (SST) over the SE Atlantic are warmer than the climatological means, although its impact on low-CF over the deployment region remains unclear at the daily to synoptic time scale.

## 1. Introduction

The southeast (SE) Atlantic and the west coast of southern Africa is one of the key regions of the globe for understanding the interactions between Earth's climate, weather, and pollution. It is characterized by a stratocumulus cloud deck associated with strong large-scale subsidence and the anticyclonic circulation associated with the semi-permanent St. Helena High above the Atlantic Ocean in the southern hemisphere (SH) (Klein and Hartman, 1993; Wood, 2015). The low-level stratocumulus clouds increase the net amount of outgoing radiation at the top of the atmosphere (TOA), inducing a negative radiative effect.

The African Easterly Jet (AEJ) dominates the mid-tropospheric circulation over West Africa from August to October. In the SH, the southern African Easterly Jet (AEJ-S), best discernible in August–November, plays a large role in modulating variations in the intensity and position of precipitation over the African continent (Nicholson and Grist, 2003; Adebiyi and



Zuidema, 2016; Dezfuli, 2017). Nicholson and Grist (2003) found that the equatorial rainbelt is approximately bounded by two jets (northern AEJ (AEJ-N) and AEJ-S) during August–November. The AEJ-S and its associated secondary circulation

through vertical motion is also an effective carrier of aerosols (Adebiyi and Zuidema, 2016). Adebiyi and Zuidema (2016) showed that about 55 percent of the biomass burning (BB) aerosols that are transported out of southern Africa during September–October are transported westward by the AEJ-S to the southern tropical Atlantic, and the remaining BB aerosols are either carried northwestward into the intertropical convergence zone or returned to southern Africa. The AEJ-S is driven by the strong meridional temperature gradients over land (Fig. 1).

Another strong low-level wind occurs around 925–950 hPa along the Namibian coast of the SE Atlantic. This is known as the "Benguela low-level jet" (Nicholson, 2010, hereafter LLJ) since peak winds ($\sim 10$ m s$^{-1}$) are observed over the Benguela current off the coast of Namibia (~10° E, 20–25° S). The LLJ is related to the strength and location of the subtropical high (Nicholson, 2010), but also generates a secondary circulation that can affect the local subsidence. LLJ in SE Atlantic has unique features contrasting from LLJ in SE Pacific along the Chilean coast; Zuidema et al. (2016) showed that

the SE Atlantic LLJ tends to be an offshore flow, helping surface divergence to depress BLH, while the Pacific LLJ is more onshore, hitting southern Peru north of the Arica Bight (~26° S, 70° W–65° W) and aiding to elevating the coastal BLH. SSTs along the Benguela coast are strongly linked to rainfall variability in the Sahel (Lamb and Peppler, 1992) and western equatorial Africa (Balas et al., 2007).

Figure 1 shows maps of the SE Atlantic region, and the elevation of the southern part of the adjacent African continent.

The distinct difference in elevation between the Congo-Zaire basin (north of 10° S) and the Namibia-Kalahari dryland (south of 10° S, 15–21° S) is shown. At mid-levels ($\sim 600$ hPa) during August–October, the strong easterly winds of the AEJ-S (black contour in Fig. 1b), transport high relative humidity (RH) plumes from the African continent over the ocean, aided in part by the topography itself (Richter and Mechoso, 2004). The mid-level AEJ-S, caused by differential latitudinal heating over the land, is also tied to the topography of Southwestern Africa around 10° S. Recirculated air by the anticyclone to the

south of the AEJ-S returns to the African continent at the south of 18S and merges with the subtropical and mid-latitude westerly jet. Precipitation associated with the Western African monsoon progresses southward over the continent during August–October. At the lower levels (at 850–925 hPa), large-scale anticyclones associated with the St. Helena High dominates off the coast of Namibia over the ocean (5–15° E, 18–35° S), with a pressure gradient to the warmer continent establishing the LLJ near the Benguela coast (Fig. 1c). The meridional potential temperature gradient and thickness

difference are large over the continent at ~10° S. This temperature gradient, associated with the heat low, the high values of thickness of geopotential height between 850 hPa and at 600 hPa over a south African plateau, is the dominant driver of the AEJ-S. After the onset of the rainy season around the end of October, precipitation reduces the local temperature gradient, and consequently weakens the AEJ-S.

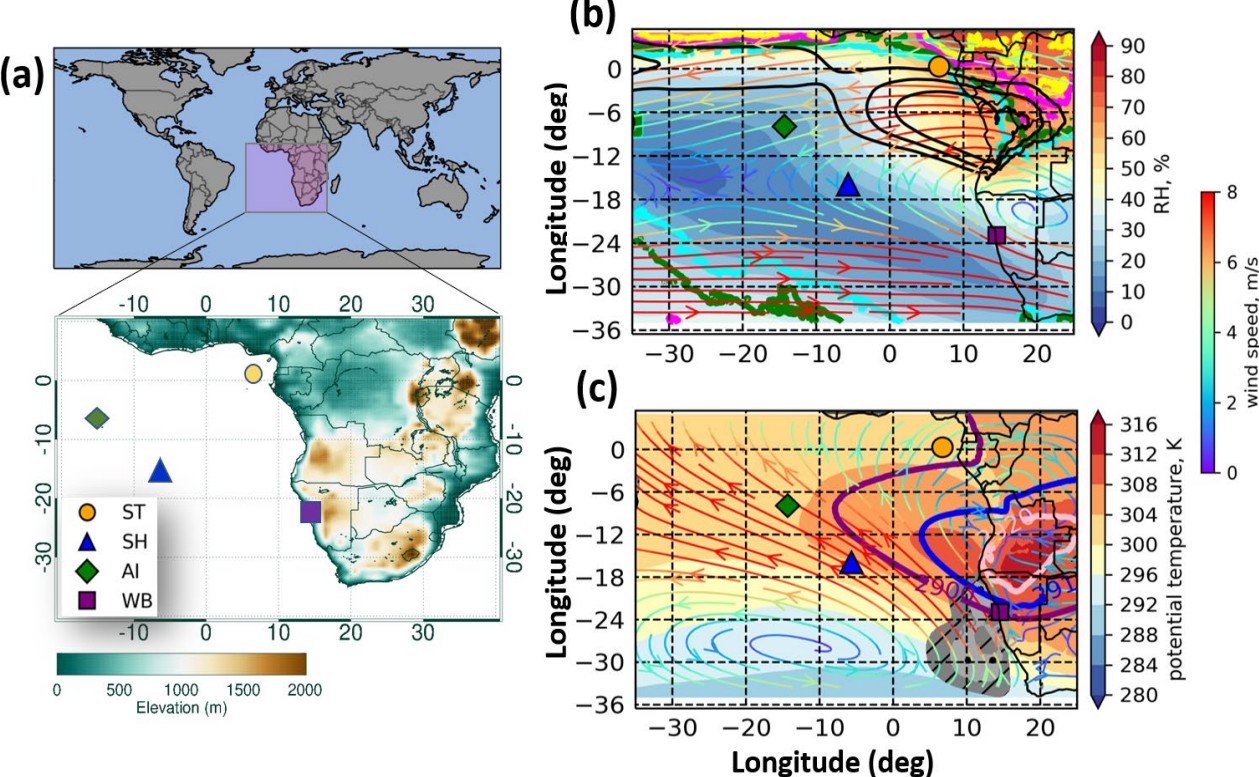

**Figure 1. (a)** Map of the ORACLES deployment region in the SE Atlantic (marked by a magenta square) and elevation of southwestern Africa. The markers represent São Tomé (ST), St. Helena Island (SH), Ascension Island (AI), and Walvis Bay (WB). **(b)** 600 hPa climatological September monthly mean (2000–2018) RH (color shading, %), overlaid by 600 hPa zonal wind speed (black contour (-6, -7, -8, m s$^{-1}$), horizontal wind streamline (m s$^{-1}$), and precipitation (yellow (0.3), magenta (0.2), green (0.1), and cyan(0.05), mm h$^{-1}$). **(c)** 850 hPa climatological September mean (2000–2018) potential temperature ($\theta$) (color shading, K) overlaid by 925 hPa horizontal wind streamline, thickness between 600 and 850 hPa (plum (2900), navy (2910), pink (2920) contours, m), and subsidence (omega at 800 hPa, gray hatched shading area (50(//), 60(.), hPa day$^{-1}$).

To characterize the interaction between aerosols and clouds, the ORACLES (ObseRvation of Aerosols above Clouds and their IntEractionS) field deployments took place during 2016–2018 over the southeastern Atlantic Ocean immediately to the west of the southern African continent. The goal was to develop an understanding of the impacts of southern African BB aerosol transport over the Atlantic Ocean on climate (Redemann et al., 2021). Collaborative international deployment



activities over the SE Atlantic such as U.K. CLARIFY[1] (September 2016, 16 August 2017–7 September 2017; Haywood et al., 2021), DOE LASIC[2] (1 June 2016–31 October 2017; Zuidema et al., 2018), and the French AEROCLO-sA[3] (22 August 2017–12 September 2017; Formenti et al., 2019) have also advanced the understanding of aerosols and their interaction with clouds. A few results from ORACLES are elucidating the observed details of aerosol-cloud interactions (Kacarab et al., 2020), the combined direct aerosol radiative effect (Cochrane et al., 2020) and the impact of moisture outflow on mid-level clouds (Adebiyi et al., 2020). However, while those studies highlight the detailed features of aerosol-cloud interaction, they cannot tell us whether the specific campaign years were typical years. Hence, understanding the meteorological characteristics during the ORACLES deployment, and how different they are compared to the climatological mean in various temporal and spatial scales is critical. This paper provide a thorough overview of the meteorology reflecting the coupled land-ocean-atmosphere system and the representativeness of the deployment months.

Aerosol-cloud interactions will be also moderated by meteorology. For example, the cloud cover changes with lower tropospheric stability (LTS), and the LTS is modified by not only surface temperature but also absorption of solar radiation by aerosols residing above the cloud over the ocean in the African region (Gordon et al., 2018; Mallet et al., 2019; 2020). Both reanalysis and regional simulations reveal that free tropospheric subsidence tends to be reduced when absorbing aerosols are present (Sakaeda et al., 2011; Adebiyi et al., 2015). Interestingly, twice more reduction in subsidence is associated with the AEJ-S than that by the BB aerosol over the ocean near the jet exit region (5–15° S, 10° W–12° E), emphasizing both AEJ-S and aerosol loadings are negatively associated with subsidence (Adebiyi and Zuidema, 2016). Large-scale subsidence contributes to establishing the boundary layer depth along with horizontal temperature advection, impacting the stratocumulus decks (Wilcox, 2010). Other studies examine how the large-scale flow influences the entrainment of smoke into the boundary layer (Diamond et al., 2018; Zhang and Zuidema, 2019; Abel et al., 2020). Thus, it is also important to identify the direct impact of the prevailing circulation on BB aerosol transport and stratocumulus decks, and to separate the meteorological impact on the stratocumulus deck from the aerosol impact on stratocumulus during the ORACLES deployment period.

The goal of this study is to describe the meteorological factors that directly impact aerosols and low clouds, particularly stratocumulus decks during the ORACLES campaign. In the next section, the dataset and methodology used are discussed. In section 3, we investigate the key meteorological features during the ORACLES deployment months compared to their climatological means. Then, we focus on the weekly-to-daily variability of the key meteorological fields during flight days to aid in the interpretation of airborne measurement in section 4. Section 5 summarizes the conclusions.

## 2. Data and methodology

---

[1] Cloud-Aerosol-Radiation Interactions and Forcing.
[2] Layered Atlantic Smoke Interactions with Clouds.
[3] AErosol RadiatiOn and CLOuds in Southern Africa.



The geographic domain of our study region is the SE Atlantic and southern Africa (30° S–5° N, 20° W–20° E) as shown in Fig. 1(a). Data and methods used to complete the relevant fields used in this study are described below.

## 2.1 Data

- Meteorological fields such as 3-D wind (u, v, ω), temperature, geopotential height (Z), specific humidity ($q$), divergence, and potential vorticity (PV) from European Centre from Medium-Range Weather Forecasts (ERA-5, Hersbach et al., 2020) are used for investigating atmospheric circulation. The analysis is based on hourly and monthly data available on a 0.25 ° longitude x 0.25 ° latitude grid with 37 vertical levels ranging from 1000 hPa to 1 hPa. The anomaly fields are computed by subtracting the climatological monthly-mean values from each monthly-mean value.

- Heat low is defined as the high values of thickness between Z at 850 hPa and Z at 600 hPa over a south African plateau. The maximum values over the plateau are larger than the 90[th] percentile of the thickness over the SE Atlantic region. We chose these levels rather than lower levels (e.g., Z700–Z925 used by Knippertz et al. (2017)) since the lower levels are often below ground in our study region of interest. The sensitivity of the strength of the heat low during the deployment to the precise choice of levels is minimal.

- The low-level tropospheric stability (LTS) is defined as the $\theta$ difference between 800 and 1000 hPa, below the aerosol layer at 700 hPa, following Adebiyi and Zuidema (2016).

- AEJ-S is defined as the horizontal wind (zonal wind at 600–700 hPa < -6 m s$^{-1}$ ) at the entrance over land and exit over the ocean within 0–10° E, 5–15° S. For August, horizonal wind at 700 hPa (600 hPa for September and October) is used.

- LLJ is defined as regions with 925 hPa horizontal wind speed in excess of 5 m s$^{-1}$) over the Benguela current off the coast of Namibia (over 0–10° E, 15–25° S).

- Microwave and infrared (MW_IR) daily Optimum Interpolation Sea Surface Temperature (OISST) from REMote Sensing Systems (REMSS) (Gentemann et al., 2004, 2010) with approximately 0.088 ° (~9 km) spatial resolution is used to characterize the interannual variability in sea surface temperatures (SST) over the Southeastern Atlantic.

- Monthly-mean Tropical Rainfall Measuring Mission (TRMM) product 3B43 with 0.25 ° grid spacing (Huffman et al. 2007) characterizes precipitation. This dataset is a combination of space-borne radar, microwave, and infrared channels with monthly calibration with surface rain gauges when available. Similar results were obtained using monthly Global Precipitation Mission data (not shown).

- Moist convection is also defined using rightness temperatures from Meteosat-10 satellite data, obtained from the NASA Langley Center. Moist convection is defined as brightness temperatures lower than 230 K.

- The Level 3 monthly cloud fraction product from the Moderate Resolution Imaging Spectroradiometer (MODIS) on board both Terra and Aqua (1° grid resolution) is used to calculate monthly mean low cloud fractions. The Level 3



daily cloud fraction product from the Visible Infrared Imaging Radiometer Suite (VIIRS, Hubanks et al., 2019) data on board the Suomi National Polar-orbiting Partnership (Suomi NPP) is used to calculate daily mean low cloud fractions.

## 2.2. Methodology for planetary boundary layer heights (BLH) estimate


The planetary boundary layer height (BLH) is estimated using 6 hourly ERA5 specific humidity ($q$) and RH based on a heuristic algorithm we have developed. The computed BLH is designed to include the decoupled cloud-topped planetary boundary layer (PBL) in which the cloud layer is above the well-mixed surface-based layer. In this quasi-isentropic exchange, the air masses mixing occurs without changing their potential temperature rapidly. This tends to be higher than the

well-mixed sub-cloud layer and provides the extent of the short-term influence of the boundary on the free atmosphere. The methodology for calculating the BLH is shown below.

(1) Calculate $dq/dz$ (i.e. vertical derivative of $q$, where $q$ is specific humidity [g kg$^{-1}$], and z is the vertical level) up to D, which is the maximum permitted BLH. Here D is 3 km over the all oceanic regions and islands south of 2° N and west of 10E. D is 6 km over land (except for the islands).

(2) Find the height of minimum $dq/dz$ at each horizontal grid point in the domains defined in (1).

(3) Find the height where $q=10$ g kg$^{-1}$ marching downward from height D. We chose q=10 g kg$^{-1}$ as a threshold because this is a reasonable value to indicate influence from the surface. The choice of $q$ value is almost invariant if we have $q$ threshold is larger than 5.

(4) Pick the higher altitude of (2) or (3).

(5) Compute a horizontal average (5 points) of the result.

Note that for examining the climatological and the monthly mean BLH variability, we used the monthly mean BLH variable from ERA5 reanalysis data instead of the BLH computed using our algorithm because this allows an examination of the typical monthly mean well-mixed layer BLH compared to the climatology during the deployment months. The computed

BLH is higher (by several hundred meters up to a few km over the continent) than the monthly mean BLH from ERA5, mainly because the computed BLH is designed to include decoupled stratocumulus above the mixed-layer. In general, spatial patterns and values near the coast south of 15° S are similar despite the differences in patterns and values over offshore north of 5° S, oceans, and land.

## 3. Seasonal mean and variability of the synoptic-scale circulation

We first examine the climatological mean and variability of the key meteorological factors directly affecting *clouds* and *aerosols* during the ORACLES deployment.

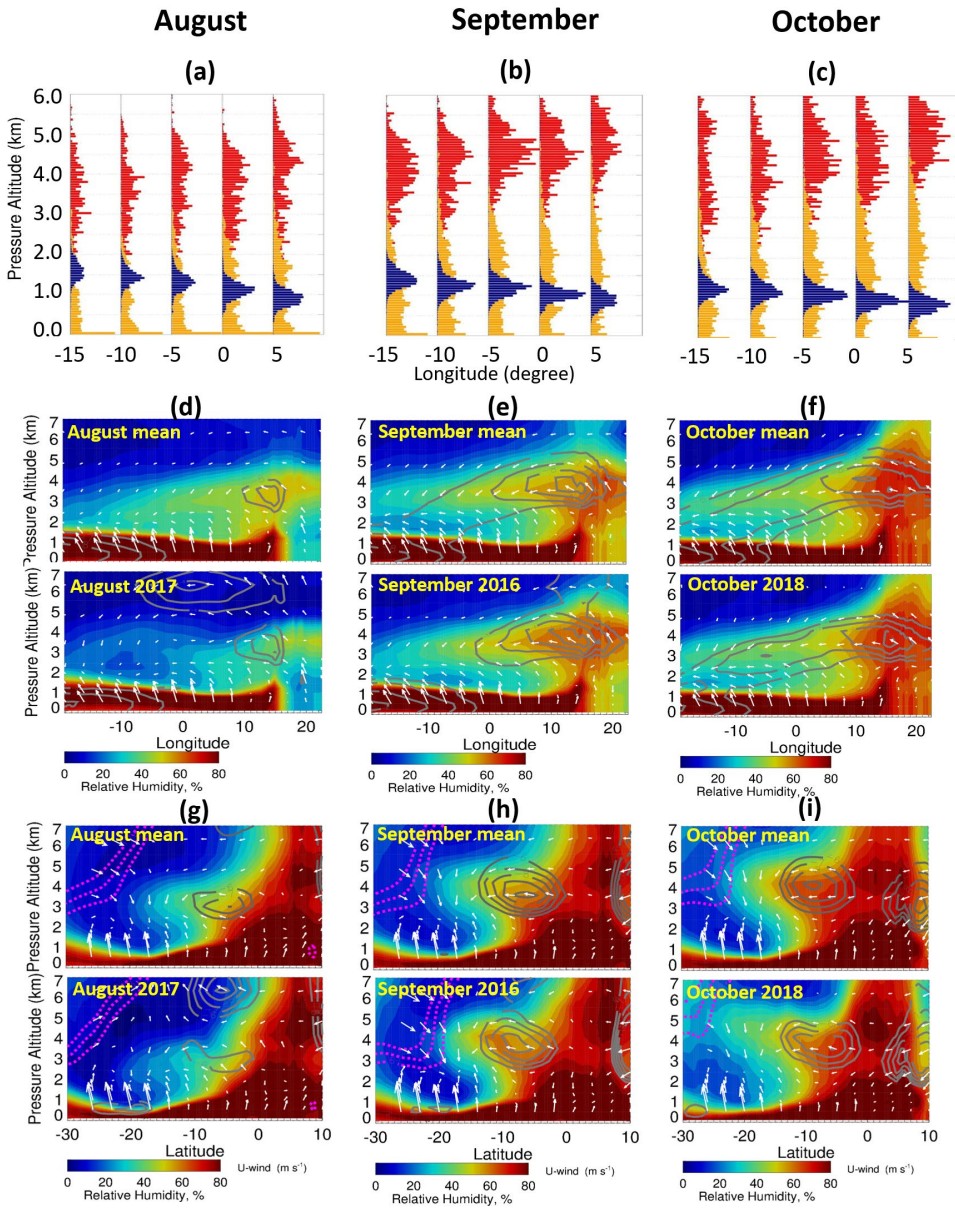

**Figure 2. (a–c)** Distributions of the climatological mean (2006–2017) aerosol layer top height (red), cloud top height (blue), and the separation distance between clouds and overlying aerosols (yellow) as a function of longitude (latitudinally averaged from 10–22.5° S) from Cloud-Aerosol Lidar with Orthogonal Polarization (CALIPSO) (Figure reproduced from Redemann et al. (2021)). **(d–f)** longitudinal cross-sections at 10° S and **(g–i)** latitudinal cross-section at 10° E of the RH overlaid by zonal wind (gray solid; easterly, and magenta dashed; westerly, m s⁻¹)





**during (top) climatological mean (2000–2018) and (bottom) the 3 years of deployment months (August 2017, September 2016, October 2018). White arrows represent horizontal wind vectors.**

The top panels of Fig. 2 show the climatological mean longitudinal cross-sections of aerosol top height, cloud top height, and their separation. Along with them, the bottom panels of Fig. 2 show longitudinal and latitudinal cross-sections of RH overlaid by AEJ-S for the climatological mean and deployment months in each of the three years. It is evident that there is month-to-month variability of the aerosol and cloud top height and their separation. The vertical extent of the aerosol layer and the depth of the separation layer between cloud and aerosols (Figs. 2(a–c)) appear to be tied to the vertical extent of

AEJ-S (Figs. 2(d–f)). That is, as the height of the AEJ-S increases from August to October, the typical aerosol layer top heights, and separation between the aerosol layer and the underlying cloud, increase as well. The aerosol top height occurs around 4.5 km averaged over SE Atlantic in September, and is similar to the heights of the highest RH (> 70 %) and maximum AEJ-S wind speed, indicating that the large-scale circulation can directly affect local aerosol fields.

The AEJ-S core is located near 4km altitude and at 8S, especially during September and October for both climatological

mean and deployment month (Figs. 2(g–i)). The enhanced RH extends up to ~ 6 km just offshore at 10° E (Figs. 2(d–f)). The southerly LLJ off the Namibian coast is also seen (~ 1–2 km, white wind vectors, Fig. 1). The climatological mean of a longitudinal and latitudinal cross-sections of RH, AEJ-S, and horizontal wind features are similar to those during the deployments. However, an additional anomalously strong upper-level jet was observed at 6.5 km (5° S–15° N) during August 2017 (Fig. 2d). This is somewhat similar to the so-called "tropospheric easterly jet" (TEJ) (Wu et al., 2009). This

TEJ is typically found at about 100–200 hPa (~ 11.7–15.8 km), 5–15° N, but its association with AEJ-S is not well understood. Furthermore, the jet altitude and latitude in August 2017 are quite a deviation from those of the TEJ. Therefore, whether this anomalous August jet is related to the TEJ is still unclear. Clearly shown is, however, that this upper-level easterly jet is one of the unique features found in August 2017 deployment, (Fig. 2g). This jet is enhanced over a relatively dry region (RH < 30 %). The mid-latitude "dry tongue" at 1–2 km penetrates into the low-level around 20–10S (Figs. 2(g–i)),

with high RH plume aloft at 3–5 km due to the AEJ-S. This dry air is tied to anomalous southerly advection of dry air originating from the southern oceans, as well as the large-scale subsidence associated with St. Helena anticyclone over the subtropical South Atlantic Ocean (Myers and Norris, 2013; Adebiyi et al., 2015). The mid-level dryness is stronger in August 2017 compared to the climatological mean (Fig. 2g). The dry intrusion along with southwesterly wind and moist plume above and south of 10S during the deployment is similar to the climatological mean in September 2016, but the dry

intrusion weakens in August 2017 and October 2018 (Figs. 2(g–i)).

## 3.1. Meteorological characteristics associated with AEJ-S



**Figure 3. (a) Map of zonal wind (black contours, m s⁻¹; values ≤ -6 m s⁻¹ with 1 m s⁻¹ interval), RH (shading, %), and horizontal wind vector at 700 hPa for August and at 600 hPa for September and October. Precipitation (line contour, light blue (0.05) to red (1.05), with 0.1 intervals, mm hr⁻¹) is overplotted for the climatological mean (2000–2018) and the ORACLES deployment months (August 2017, September 2016, and October 2018). The color boxes (red, blue, and green) indicate the month of ORACLES deployment. (b) Precipitation anomaly averaged over magenta dashed box region (5–20° E, 5° S–20° S, in top left panel of (a), mm hr⁻¹) for August– October in 2016–2018. The blue, red,**



**and green arrow denotes the value in September 2016, August 2017, and October 2018 ORACLES deployment, respectively.**

The AEJ-S and RH characteristics during the individual deployment months (August, September, and October) are shown in Fig. 3. Since the maximum core of AEJ-S in August is lower than in September and October as shown in Fig. 2, wind and RH at 700 hPa (~ 3 km) is plotted for August. Clearly apparent are the southward "march" of the regions of significant RH and continental rainfall, and strengthening of the easterly jet from August to October. The AEJ-S exhibits substantial month to month and year to year variability. In general, the AEJ-S is relatively strong during September and October and weaker in August. The AEJ-S maximum wind speed is largest in September, confined over the coastal region over 5–15° S. The maximum wind speed is weaker in October than that in September, but the jet extends more westward over the tropical Atlantic Ocean. Note that the "recirculation" pattern (shown by reversing wind vectors around 15° W–10° E, 10–25° S) is present for all three months but is only about half as strong in August as in the other two months. The zonal extent of the recirculation pattern appears to be associated with the strength of AEJ-S at 600 hPa: the stronger the AEJ-S, the larger the radius of recirculation.

The AEJ-S in August 2017 is weaker than the climatological mean. The region around 5–10° S, 0–15° E is slightly drier in August 2017 than the August climatology, but precipitation over the continent penetrates further south than the climatology. In September 2016, the 600 hPa RH is higher to the southwest of the AEJ-S compared to climatology, but the difference is relatively small. September 2016 is wetter than climatological mean with higher RH offshore (Fig. 3b). Inland precipitation is also pronounced around 5° N, penetrating further south from August to October. In October 2018, The AEJ-S jet strength is also slightly weaker than climatology, but the difference is small (less than ~ 1 m s$^{-1}$). October 2018 around 5–20° S, 5–20° E is also wetter than the climatological mean. The distribution of RH, which is tied to the southward extent of precipitation, varies from year to year.




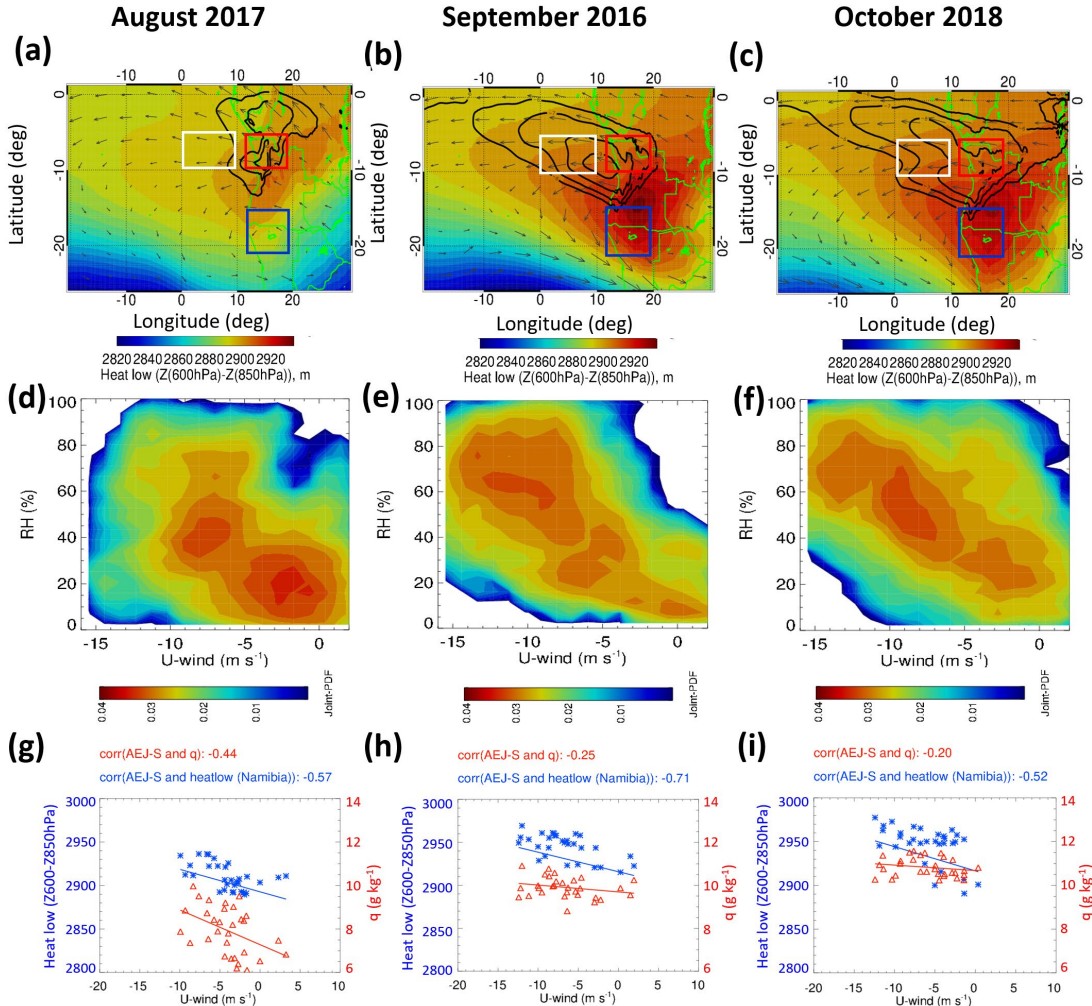

**Figure 4. (a)** Map of thickness (geopotential height (**Z**) difference between 600 hPa and 850 hPa) overlaid by the zonal wind (contour, black, m s$^{-1}$) and wind vector at 700 hPa in August 2017. **(b, c)** the same as **(a)** except for 600 hPa in September 2016 and October 2018. **(d)** 2-D Joint-probability density function (pdf) of zonal wind at 700 hPa and RH at 700 hPa in August 2017 averaged over the jet exit region (white boxed region: 0–10° E, 5–10° S). **(e, f)** the same as **(d)** except for 600 hPa in September 2016 and October 2018. **(g–i)** The scatter plot of zonal wind at 700 hPa (white box region: 0–10° E, 5–10° S, m s$^{-1}$) and the heat low (blue box region: 12–18° E, 15–21° S, m) for August 2017.

The red triangles denote the scatter plots of zonal wind at 700 hPa (white box region: 0–10° E, 5–10° S, m s$^{-1}$) and specific humidity (**q**) at 850 hPa (red box region: 12–18° E, 5–10° S, g kg$^{-1}$) for August 2017. **(h, i)** the same as **(g)** except for 600 hPa zonal wind in September and October. The 18 UTC and 6 hourly data are used for **(a–c, g–i)** and **(d–f)**, respectively. The boxed regions are shown in **(a–c)**.



To better understand how AEJ-S is associated with RH, the map of the heat low over the continent is examined in Fig. 4. The strongest heat low is shown in September and October, centered over 12–20° E between 10–20° S. The heat low is found

over the Namib-Kalahari dryland (12–18° E, 15–21° S), while the AEJ-S is observed near the border between Namibia-Kalahari dryland and Congo-Zaire basin (5–10° S). AEJ-S and heat low are closely associated through their strengths and meridional extents. For example, both AEJ-S and heat low are strongest in September 2016, while they are weaker but most extensive, expanding to the ocean, in October 2018. The heat low is weaker in August, consistent with the thermal wind relationship, in which a stronger horizontal temperature gradient leads to a stronger vertical wind shear, leading in turn to a

stronger AEJ-S.

Joint pdfs between the zonal wind at 600–700 hPa and RH (0–10° E, 5–10° S) indicate that RH and AEJ-S relationships in September and October are very similar, with a stronger jet advecting more moisture than a weaker jet. During August 2017, the AEJ-S is generally weak under dry conditions (RH < 20 %, Fig. 4d). In contrast, the AEJ-S is strong when RH was high (> 60–70 %) in September and October (Figs. 4(e, f)). When AEJ-S is weak, September is much drier than in

October (Figs. 4(e–f)). This can be also explained by moist convection over the continent creeping southward from September to October. The scatter plots show that the AEJ-S–heat low correlation is strongest in September (Figs. 4(a–c)). Another noteworthy point is that the correlation (R) between AEJ-S and heat low in October (August) is approximately~ -0.52 (-0.57), corresponding to an explained variance of only ~25 % (Fig. 4i). This suggests that other factors may affect the strength of the jet besides the thermal wind relation at daily time scales. The correlation using daily averaged data show

similar results and slightly lower correlations (not shown). The correlation between AEJ-S and inland $q$ is small, indicating that inland moisture itself may be less likely to affect the formation and maintenance of the AEJ-S. This can be understood in the same context as Jackson et al. (2009), in which the AEJ-S may enhance convection through enhancing the vertical ascent at the jet entrance region, with the additional advected moisture acting to reduce the thermal contrast (Adebiyi and Zuidema, 2016). Nonetheless, the question still remains whether the AEJ-S itself may result from convection, because some studies

claimed that latent heat release is a factor in the development of the AEJ-N (Thorncroft and Blackburn, 1999).

**Figure 5. (a–c)** Map of omega at 800 hPa (subsidence, shading, hPa day$^{-1}$) overlaid by thickness between 600 hPa and 850 hPa (color contour with 10 m interval from 2900, m) and zonal-wind at 700 hPa (black contour, m s$^{-1}$) and horizontal wind at 700 hPa (wind vector, m s$^{-1}$) in **(a)** August for (top) the climatological mean and (bottom) the deployment month. **(b, c)** The same as August except for 600 hPa in September and October. **(d–f)** Latitudinal cross-section of vertical motion (omega (ω), shading, hPa day$^{-1}$) averaged over the jet entrance region (12–20° E) overlaid by wind vector (vector, meridional wind and -1*omega (v, ω), m s$^{-1}$; for positive value to represent the ascent) and zonal wind (black contour, m s$^{-1}$) for (top) the climatological mean and (bottom) the deployment month. The gray filled area represents the inland topography. The red asterisk in **(d–f)** represents the AEJ-S core.



While Fig. 4 suggests that the latitudinal temperature gradient is correlated to AEJ-S, how it maintains the AEJ-S is not clear. We further look at the vertical motion along with the heat low and other meteorological variables during the deployment months in Fig. 5. The notable features are: (1) strong upward motion in the jet entrance region inland (10–15° S, 12~20° E) in September 2016 and October 2018 (Fig. 5(b–c)); and (2) an updraft found over the region (5–10° S, 15–25° E) associated with AEJ-S in August 2017 (Fig. 5a).

The AEJ-S core is located north (0–5° S; Fig. 5a, d) and lower in altitude (~3km; Fig. 5d) in August compared to September and October (5–10° S; ~ 4 km (Fig. 2, Figs. 5(b–c, e–f)), and the jet cores are tied to the strength of the inland heat low (Figs. 5(a–c)). The maximum AEJ-S core is found around 3 km (~700 hPa) in August, explaining why the jet strength at around 4 km (~600 hPa) is weaker in August than in September and October (Figs, 2. 5). The overall strength of AEJ-S in August is weaker than those in September and October (Figs. 5(a–c)). Furthermore, the strong upward motion inland appears to be associated with stronger AEJ-S in the SE Atlantic coast (Figs. 5(d–f)). This may also contribute to the aerosol transport efficiency, supported by the finding of Adebiyi and Zuidema (2016), which showed that the AEJ-S also contributes to the efficiency of aerosol lofting through an ascent at the AEJ-S entrance region (10–15° S, 12–18° E). This ascent seems to also be associated with a large heat low centered between the Congo-Zaire basin (~5–10° S,~15° E) and Namibia-Kalahari dryland (~15-21° S, ~15° E) (Figs. 5(a–c)), which is supported by a convergence of surface winds (Figs. 5(d–f)). The low-level convergence is strongest in September (not shown) and the inland ascent is also largest in September as well (Fig. 5e). The inland upward motion (ω) seems to also be associated with the converging low-level surface winds, which is tied to mid-level divergence (around 600 hPa) of the ageostrophic wind, also given by Kuete et al. (2020) (not shown). This ageostrophic motion through the secondary circulation is important in that it is associated with the vertical motion, which drives the jet in the entrance region.

Large-scale subsidence over the SE Atlantic, especially over the subtropical region (15– 25° S) appears to also be associated with the strength of AEJ-S (Figs. 5(a–c)). The weaker AEJ-S is associated with strong subsidence, while the stronger AEJ-S is associated with the suppressed subsidence, and this is more clearly shown in August and September (Figs. 5(d–e)). This result is consistent with finding by Adebiyi and Zuidema (2016) that AEJ-S induces an upward motion below the AEJ-S, reducing the mean subsidence over the ocean north of 20S, while strengthening the large-scale ascent over the continent, including over aerosol source regions. Compared to the climatology, subsidence is slightly stronger in August 2017 and September 2016, and slightly weaker off the coast (south of 20° S) but slightly stronger over the land in October 2018 than the climatology (Figs. 5(d–f)). Figure 5 summarizes the key feature of the AEJ-S: enhanced ascent and a stronger heat low accompanied by surface wind convergence near the jet entrance region, and the weak subsidence over the SE Atlantic. For more detailed feature during all August, September, October in 2016–2018, see Fig. 1S in the supplementary material.

### 3.2 Meteorological characteristics associated with low-level cloud



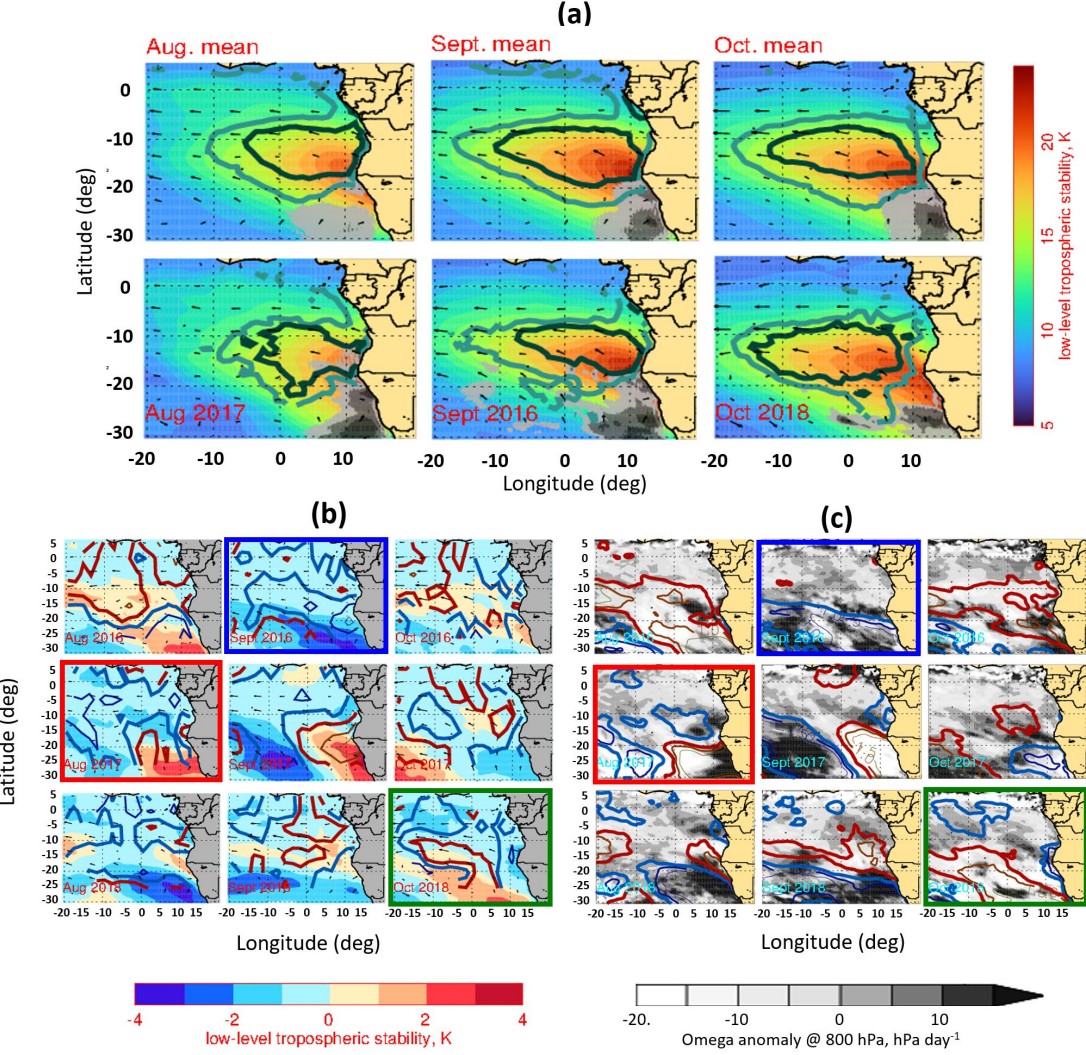

**Figure 6.** Longitude-latitude cross-section of monthly mean (a) LTS (shading, color, K) and low-level cloud fraction (low-CF, cloud-top height below 2.5 km, 0.8 (light green) and 0.9 (dark green), %) overlaid by subsidence (omega at 800hPa, 50, 65, 80 hPa day$^{-1}$, increasing from the light gray to dark gray) and horizontal wind vector (black) at 800 hPa for (top) the climatological mean (2003–2018 for low-CF and 2000–2018 for other variables) and (bottom) the deployment month. Monthly mean (b) LTS anomaly (color shading, K) overlaid by low-CF anomaly (contours: blue (-0.02, -0.07, -0.12 (from thick to thin)), and dark red (+0.02, +0.07, +0.12 (from thick to thin)), %), and (c) vertical velocity (ω) at 800 hPa anomaly (gray shading, hPa day$^{-1}$) overlaid by LTS anomaly (contours: blue (-0.5, -1.0, -1.5 (from thick to thin)), and dark red (+0.5, +1.0, +1.5 (from thick to thin)), K). The low-CF anomaly is smoothed by



**averaging at 2° (longitude) by 5 ° (latitude) to reduce the noise. The color boxes (red, blue, and green) indicate the month of ORACLES deployment.**

The stratocumulus deck, which accounts for most of the low cloud over the SE Atlantic, is affected by LTS and large-scale subsidence (Wood and Bretherton, 2006; Wood et al., 2015; Der Dussen et al., 2016; Fuchs et al., 2018; Adebiyi and Zuidema, 2018). To examine the impact of large-scale circulation on low cloud during the ORACLES deployment months, we first consider: 1) the relationship between the low-level cloud fraction (low-CF, cloud-top height below 2.5 km), large-scale subsidence, and LTS in Fig. 6; and 2) that between low-CF, LLJ, and BLH in Fig. 7. For all months, LTS is positively
correlated with low-CF (Fig. 6b). The LTS in August 2017 and October 2018 near the south of offshore Namibia is higher than the climatology and low-CF is higher there too. Furthermore, there seems to be quite a strong correlation between the subsidence anomalies (positive ω anomalies) and the negative LTS anomalies, especially in August and September (e.g. August 2018, September 2016, 2017, 2018). However, exceptions are found in August 2016, August 2017, October 2017, and October 2018, especially off the Namibian coast south of 20° S and over the jet exit region (5° W–10° E, 5–15° S).

The strong subsidence tends to reduce low-CF, but this association appears to vary with regions and months. In September, the low-CF appears to increase when the large-scale subsidence is reduced offshore (e.g. September 2016, 2017 in Fig. 6c). For example, the large-scale subsidence off the Namibian coast (~13° E, ~22° S) is stronger by about 10–30 hPa day$^{-1}$ compared to the climatological mean, and apparently linked to reduced low-CF in September 2016. Both August 2017 and September 2016 have reduced low CF, but with opposite subsidence anomalies (i.e. August 2017 has anomalously low
subsidence and September 2016 has anomalously high subsidence). Low-CF during October 2018 is slightly higher than the climatological mean over SE Atlantic, but lower off the coast. For this month, the regional distributions of low-CF anomaly and subsidence anomaly suggest a positive correlation between subsidence anomaly and low-CF anomaly, especially off the coast of southwestern African (5–12° E, 0–25° S, Fig. 7e, October 2018). Thus, the subsidence–low-CF relationship is also not very clear because 1) subsidence can indirectly enhance clouds by enhancing inversion strength or reduce clouds at the
given inversion strength (Myers and Norris, 2013) and 2) the response of low clouds to vertical velocity differs as a function of timescale (De Szoeke et al., 2016; Adebiyi and Zuidema, 2018).


**Figure 7. Map of monthly mean BLH (color shading, m), the horizontal wind (vector, m s⁻¹) and wind speed (magenta contours, thick lines (> 9 m s⁻¹)) at 925 hPa overlaid by low-CF as in Fig. 6 for (a) the climatological mean, and (b) the deployment for August 2017, September 2016, and October 2018. Map of monthly mean (c) wind speed anomaly at 925 hPa (color shading, m s⁻¹) overlaid by low-CF anomaly (contours: blue (-0.02, -0.07, -0.12 (from thick to thin)), and dark red (+0.02, +0.07, +0.12 (from thick to thin)), %), (d) BLH anomaly (color shading, m) overlaid by low-CF anomaly (contours as in (c)), and (e) vertical velocity (ω) anomaly at 800 hPa (gray shading, hPa day⁻¹) overlaid by low-CF anomaly. The low-CF anomaly is smoothed by averaging at 2° (longitude) by 5° (latitude) to reduce the noise.**





The most important features of the BLH in Fig. 7 are: (1) an overall decrease in BLH from August to October; and (2) a change from high BLH over the ocean to low BLH near the coast (Figs. 7(a, b)). South of ~10° S, the boundary layer (BL) flow is related to the strength of the St. Helena high-pressure system (centered around the 30° S and 20° W), implying BLH can be tied to the strength of the large-scale subsidence and that of the zonal advection of warm mid-tropospheric air. Indeed, the increased subsidence tends to be tied to shallow BLH, as shown near the Namibian coast (5–10° E, 20–25° S) in August

2017 deployment and near the southeastern ocean (0–10° E and 25–30° S) in October 2018 deployment (Fig. 6c, right panel of Fig. 7d).

Another interesting finding is that the strength of the LLJ increases from August to October, especially off the coast of Namibia around 15–25° S, 0–10° E (Fig 7a). Furthermore, the LLJ is strong when subsidence off the Namibian coast is strong (Fig.6c and Fig. 7c) in September. Then, a question arises here: can we expect the strong AEJ-S, by reducing

subsidence, to generate a deeper BLH? Intuitively, one might expect reduced subsidence to lead to a deeper stratocumulus-topped boundary layer, but this is not always the case when the LLJ is weaker (Fig. 7c, October 2018) or nonlinear processes such as mixing due to entrainment (Mazzitelli et al., 2014). Zonal advection of warm temperatures via a larger-scale recirculation pattern has also been shown to dominate (Adebiyi et al., 2015). The LLJ in both August 2017 and September 2016 is enhanced relative to climatology, but the LLJ is suppressed in October 2018 compared to the climatology (Fig. 7c).

Reduced (enhanced) low-CF tends to be associated with high (low) BLH (Fig. 7d), but also shows large spatial variability. For example, September 2016 has generally anomalously high BLH, but there is anomalously strong subsidence in most of the region. The high low-CF tends to be tied to LLJ, but they show large spatial variability, especially over 25–30° S, 15–20° W (Fig. 7c). For instance, in October 2018 both strong AEJ-S and strong LLJ tends to be tied to reduced low-CF, and in August and September strong LLJ is tied to the reduced low-CF, but this relationship does not hold in all years (Fig. 7b).

Both AEJ-S–LLJ and low-CF–LLJ relationship over SE Atlantic is highest in October 2018, which is likely associated with developing mid-latitude frontal system. As discussed in Fig 6, subsidence–low-CF relationship varies with regions and months (Fig 7e). Thus, the relationship among AEJ-S, LLJ, subsidence, and low-CF is not simple. We will delve this into more in detail during the individual deployment month in section 4. All the maps of monthly mean and anomaly of subsidence, LTS, BLH, and LLJ with low-CF anomaly for August, September, and October during 2016–2018 are present in

Figs. 2S–4S in the supplementary materials.



**Figure 8.** Map of (a) monthly mean of SST overlaid by the mean sea level pressure (SLP, color contours), and (b) SST anomaly from the climatological mean (2002–2018) overlaid by low-CF anomaly (contours: blue (-0.02, -0.07, -0.12 (from thick to thin)), and dark red (+0.02, +0.07, +0.12 (from thick to thin)), %) and SLP (thin color contours). (c) Time series of SST anomaly averaged over box region (18° W–15° E, 5–20° S) for the ORACLES deployment months (August 2017, September 2016, and October 2018). The markers in (a, b) represents São Tomé (magenta triangle), St. Helena Island (red cross), Ascension Island (green asterisk), and Walvis Bay (light blue diamond). The SST anomaly is averaged at ~2.11°, and the low-CF anomaly is smoothed by averaging at 2° (longitude) by 5° (latitude) to reduce the noise.



How the low-CF varies in association with the SST and SLP during the deployment years is examined in Fig. 8. The
SST over the SE Atlantic tends to be slightly warmer than the climatological mean for all years, (Fig. 8c). Warmer SSTs tend
to reduce the stratocumulus cloud fraction, by reducing the static stability (Wood et al., 2015). Warmer SST seems to be also
associated with slightly more rainfall over SE Atlantic regions (see Fig. 3), consistent with Knippertz et al. (2017). August
2017 has the warmest SST anomaly in the 10–20° S region, and the smallest CF fraction of the years considered. The St.
Helena High is the weakest in October 2018 compared to the other two deployment months. (The monthly mean SST and
SLP for all August, September, and October during 2016–2018 is shown in Fig. 5S in the supplementary materials).
However, this relationship between regional anomalies of SST and low-CF does not hold in September 2016. For example,
reduced low-CF is found when cool SST anomaly occurs in September 2016 (Fig. 8b). The robust correlation between SST
and low-CF at the daily time scale is also not obtained for October 2018, especially over the coastal region, although we
notice the warmer SST is linked to the decreased low-CF anomaly especially over the tropical Atlantic (5° N–10° S, 20° W–
0). Warming is more intense over the tropical Atlantic, although the warming trend is still reflected over the ORACLES
flight region (5–20° S, a boxed region in Fig. 8b).

Moisture transport, either through entrainment or radiation, can also alter low cloud behavior. However, it seems that
the simple SST – low-CF relationship is not achieved without considering 1) interaction among subsidence, LTS, BLH, and
LLJ and 2) time-scale of the variability in SSTs (Klein et al., 1995; De Szoeke et al., 2016). This will require a more focused
study.

## 4. Variability of synoptic-scale circulation during the deployment

In this section, we will investigate the variability of the synoptic-scale circulation during the 3 ORACLES deployment
months, including detailed analysis of the selected flight cases.

**4.1 Deployment year 1 (Namibia, September 2016)**

The first deployment of ORACLES was based in Walvis Bay on the Namibian coast of southwestern Africa. Characteristics
of synoptic-scale and convective features during the September 2016 deployment are summarized in Table 1.

**Table 1. Characteristics of synoptic-scale features over SE Atlantic during the September 2016 deployment.**

| Dates | Flight days | Focused Lon/Lat | Synoptic description |
|---|---|---|---|
| 31 Aug.– 2 Sept. | 31 Aug., 2 Sept. | 10° W–20° E, 0–10° S | Fast-moving oceanic moisture disturbance, with very weak AEJ-S |
| 2–6 Sept. | 2, 4 Sept. | 10° W–15° E, 0–15° S | No moisture advection from land along with no AEJ-S signature |





| 6–9 Sept. | 6, 8 Sept. | 20° W–15° E, 0–15° S | Weak moisture advection from land along with the emerging AEJ-S |
|---|---|---|---|
| 9–13 Sept. | 10, 12 Sept. | 20° W–20° E, 0–15° S | Relatively strong moisture advection from land along with the strengthening of the AEJ-S. positive vorticity advection associated with St. Helena High (anticyclones) developments. Moderate diurnal variation of RH, horizontal wind, potential temperature ($\theta$), and heat low. |
| 13–16 Sept. | 14, 16 Sept. |  | Fast moisture advection (13 m s$^{-1}$) from land along with the strong enhancement of AEJ-S, Mesoscale Convective System. |
| 16–21 Sept. | 18, 20 Sept. | 10° W–10° E, 5–25° S | Second fast (~8 m s$^{-1}$) moisture advection along with developing AEJ-S and emerging LLJ |
| 21–26 Sept. | 22, 24, 25 Sept. | 10° W–20° E, 5–25° S | Relatively weak subsidence near the coast of Namibia. Strong AEJ-S. |
| 28–29 Sept. |  |  | Suppressed moisture transport along with the weakening of the AEJ-S |





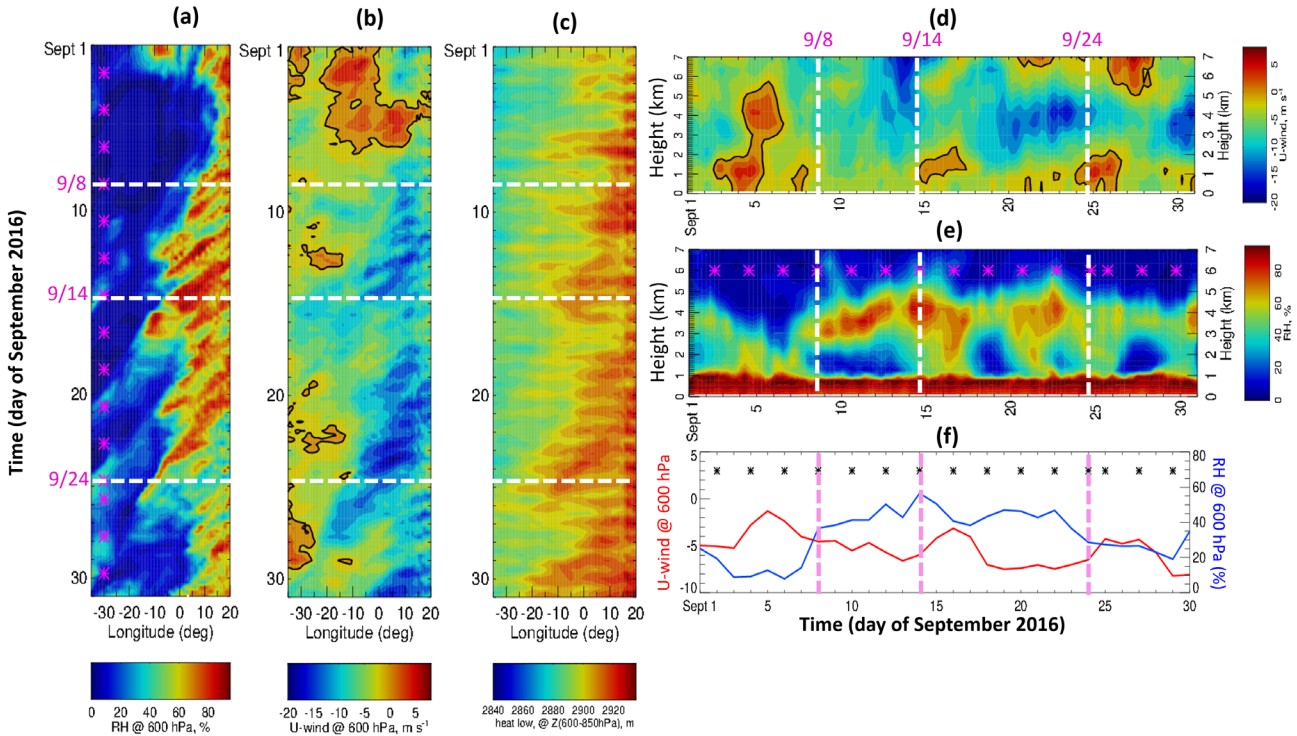


**Figure 9. Longitude-time cross-section of: (a) RH at 600 hPa (shading, %), (b) zonal wind at 600 hPa (shading, m s⁻¹), (c) heat low, and (d–e) altitude-time cross-section at 10° E, averaged over 8–10° S during September 2016. The black contour in (b) and (d) represents 0 value. The white dashed lines indicate the flight days investigated further in this study, and the asterisks represent the flight days during September 2016 deployment. (f) Time series of the zonal-wind at 600 hPa (red line) and RH at 600hPa (blue line) averaged over 0–10° E and 6–10° S.**

Figure 9 shows the Hovmöller diagrams of RH and zonal wind at 600 hPa, and heat low averaged over 8–10° S for September 2016. The AEJ-S is strongly associated with the continental heat low during September, consistent with Figs. 4 and 5. Variability ranging from diurnal to weekly time scales was shown in all variables, but the diurnal heat low variability was one of the significant factors to drive the variability of the AEJ-S. Interestingly, the increased AEJ-S along with enhanced RH is shown on 8–15 September 2016 between 10° W and 20° E, transporting moisture from continent to ocean, but the AEJ-S leads RH by 0–2 days during that period (Figs. 9(d, e, f)). This suggests the moist transport from the land can be facilitated by the westward propagation of the AEJ-S (Figs. 9(d–f)).



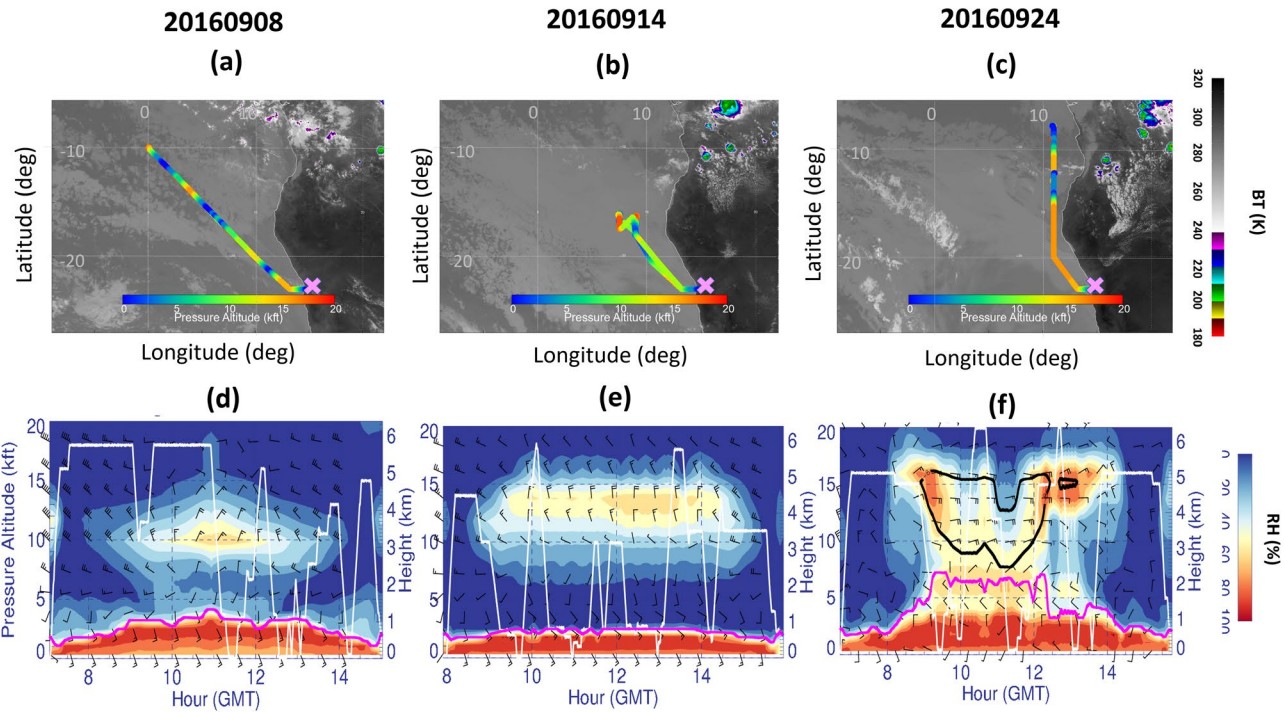

**Figure 10. (a–c) The horizontal flight tracks during September 2016 ORACLES deployment plotted on the Meteosat IR 10.8 μm imagery at 1345 UTC. The color represents the altitude of the flight along the horizontal flight track. (d–f) Curtain plot of RH along the flight track during 8, 14, and 24 September 2016. The white contour represents the flight profile. The magenta line in (d–f) represents the BLH along the flight track. Bold black contours in (d–f) are zonal wind ( -8 m s$^{-1}$).**

Figure 10 shows three ORACLES flight days during September 2016. The flights on 8 and 14 September 2016 were so-called "routine" flights, designed to develop reasonable statistics along a particular track. The flight on 24 September 2016 was a targeted flight to reach as far north as possible through the African smoke plume (Redemann et al, 2021). These cases were selected based on their unique meteorological and aerosol features. While September 8 and 14 show relatively dry conditions (RH < 50%) throughout the flight track, September 24 has overall moist conditions especially at the high altitude ranging between 12– 17 kft (3.6 ~ 5.1 km), especially around 12–14° S. Note that the routine flight on September 8 was dry despite the discussion of Fig. 9 because the moist plume was north of 10° S (e.g., 5–10° S).

The consistent feature during September 2016 is the moist plumes with RH ~ 50–60 % in the range of 10–18 kft (~3– 5.5 km) intercepted by the flight tracks (Figs. 10 (d–f)). Especially during the September 24 flight, the peak RH lines are aligned with the AEJ-S (black contour, Fig. 10f), where its maximum is observed around 5–10° S, 10–20° E, originating



from the continent. Furthermore, high RH is observed throughout the vertical layers from the bottom, indicative of a slightly
deeper marine boundary layer compared with the other two highlighted flight days (Fig. 10f). Note that the moisture along
the flight track in Fig. 10 also largely depends on where the flight track is with respect to the moisture plume.

**Figure 11.** Map of (a–c) specific humidity ($q$) at 600 hPa (shading, g kg$^{-1}$) and horizontal winds at 600 hPa (vectors, m
s$^{-1}$) overlaid by thickness (geopotential height ($Z$) difference between 600 hPa and 850 hPa) (color contour >2920 m,
10 m interval; high values over land represents the heat low). The black line represents the horizontal flight track on
the given day. Map of (d–f) potential temperature ($\theta$) at 850 hPa (shading, K) and horizontal winds at 925 hPa





**(vectors, m s⁻¹) overlaid by Z at 1000 hPa (blue contour, m) at 1200 UTC 8, 14, and 24 September 2016. (g) Time series of daily averaged (top) vertical velocity (ω) at 800 hPa (orange) and low-CF (blue) over region B, and (bottom) AEJ-S wind speed (red: horizontal wind at 600 hPa over 0–10° E, 5–15° S, region A) and LLJ wind speed (green:**
**horizontal wind at 925 hPa over 0–10° E, 15–25° S, region B) during September 2016. The asterisk in (g) represents the September 2016 flight days. The magenta line refers to the three flight days. (h) (h) 2-D joint pdf of the daily averaged AEJ-S wind speed and LLJ wind speed (correlation is obtained for 5–30 September 2016 (blue dots) when the AEJ-S develops). The marginal plot shows their normalized pdf for whole month (gray) and the days during the month when AEJ-S develops (red), respectively.**

The moisture transport by the AEJ-S has a different pattern for each of the flight days, as shown in Figs. 11(a–c). On 8 September 2016, there are large moisture gradients over the ocean around 5° W–10° E, 10–20° S accompanied by a relatively weak AEJ-S associated with a midlatitude cyclonic circulation expanding northward up to 20° S (Fig. 11a). On 14 September 2016, the stronger AEJ-S transports moisture over the ocean at 10S, but the jet also gets weaker (<10° W, 10–15° S) as subtropical cyclonic circulation develops south of 20° S (Fig. 11b). As the mid-latitude circulation gets weaker and

confined west of 10W and south of 20° S, the AEJ-S extends further northwest on 24 September 2016 around < 10° W, 0–10° S (Fig. 11c). The large horizontal $\theta$ gradient between the Congo-Zaire-basin and the Namibia-Kalahari dryland is observed for all flight days, just south of where the maximum AEJ-S is observed.

   AEJ-S appears to be weakly associated with LLJ in September 2016 (Fig. 11g), especially after AEJ-S develops (at around 5–6 September 2016). The joint pdf also shows that AEJ-S is weakly associated with LLJ in September when we

consider after AEJ-S emerges at the beginning of September (Fig. 11f). AEJ-S–LLJ relationship may be explained by the linking factor, such as subsidence, which can influence both AEJ-S and LLJ at different time scales. For example, AEJ-S–LLJ tends to behave in opposite direction depending on the strength of subsidence (e.g. 6, 16–17 September 2016; Fig. 11g). However, this relationship varies with regions and in different time scales, indicating that multi-year data analysis will be needed to confirm the relationship between AEJ-S and LLJ.




**Figure 12. (a–c)** *Ocean*: **latitudinal cross-section of RH (shading, %), horizontal winds (wind barbs, green, m s⁻¹), θ (navy, K), and BLH (magenta, m) at 0° E (top) and skew-T log-P diagram averaged over 0° E and 9–10° S (bottom: from the left) at 1200 UTC 8, 14, and 24 September 2016. Bold black contours are zonal wind (-8 m s⁻¹). (d–f)** *Land*:



**the same as in (a–c) except for the cross-section at 18° E and skew-T log-P diagram averaged over 18° E and 5–6° S (CZ: Congo-Zaire basin) and 18° E and 16–17° S (NK: Namibian-Kalahari dryland). The gray filled area represents the topography. The vertical lines in the cross-section plots refer to the latitude we examine.**

Convection type (dry convection or moist convection) can affect aerosol impact on clouds since moist contents can alter aerosol properties (Tao et al., 2012). In order to differentiate dry from moist convection, soundings and longitudinal cross-
sections of RH, winds, and $\theta$ over the continent and the ocean are shown in Fig. 12. Over the ocean, moist plumes at 3–5 km with relatively high RH (> 60 %) south of 10° S is associated with the moist plume transported from south of 10S over the continental region (Figs. 12(a–c)). The oceanic soundings show the moist marine boundary layer topped by RH of 100 % (shown by dew point temperature, estimated using 3-D temperature and RH (Alduchov and Eskridge, 1996), is close to the current air temperature) at around 900 hPa, Figs. 12(a–c)), which represents the stratocumulus. Moisture transported from
the continent by the AEJ-S near 600 hPa produces enhanced RH; saturation occurs occasionally, producing scattered middle clouds (as on September 14 (Fig. 9b) at 11° E and 12° S).

Over the land, dry convection is dominant south of 10° S. This is confirmed by 1) air temperature follows the dry adiabatic lapse rate up to 600 hPa, and 2) dew point temperature follows the constant water vapor mixing ratio and gets close to air temperature, favoring high RH but still maintaining dry condition below 600 hPa, indicating the formation of dry
convection.. (Figs. 12(d–f)). The dry convection can reach up to 500 hPa over the Namibian-Kalahari dryland. There is insufficient moisture for rainfall, but RH reaches higher than 60% and clouds can form near the top of dry convective regions (Adebiyi et al., 2020). This is because the temperature decreases rapidly with altitude in an environment of constant (well-mixed) water mixing ratio and near-constant water saturation temperature.  The AEJ-S is at or north of the region of dry convection.

North of 10° S, the sounding profiles indicate the presence of moist convection (upper panels (18° E, 5–6° S (CZ)) of Figs. 12(d–f)). In fact, all three soundings at 5–6° S have significant Convective Available Potential Energy (CAPE), which is a measure of the amount of energy available for convection. This is consistent with Fig. 3, showing precipitation north of about 10° S. These soundings also show a stable layer above 500–600 hPa, overlaid by dry air to 300 hPa. South of 10° S (lower panels (18° E and 16–17° S (NK)) of Figs. 12(d–f)), significant moisture transport does not occur above 500 hPa. The
dry convection in the soundings over dry/hot Namibian-Kalahari dryland (16–17° S) extend up to 500 hPa (~5.5 km), especially on September 24. This results in near-saturated conditions at the top of the deep boundary layer (Fig. 12f). This, in turn, explains the moist plume that is transported westward at 15–20° S at high altitude (Fig. 12c) compared to other cases (Figs. 12(a, b)). For all cases, the moisture plume around 600 hPa over the ocean is coming from the AEJ-S and originating from the land.




**20160908**          **20160914**          **20160924**

**Figure 13.** Map of (a–c) low-CF (shading, %) overlaid by LTS (contour, K) and horizontal winds at 925 hPa (black vectors, m s$^{-1}$), (d–f) BLH (shading) overlaid by horizontal winds at 925 hPa (vectors, black), and (g) (top) daily time series of low-CF (blue lines, %) and LTS (red lines, K), and (bottom) low-CF (blue line, %) and BLH (magenta lines, meter) during September 2016. The solid boxes in (a) are averaged over region A (magenta box, 0–10° E, 5–15° S), and dashed lines are averaged over region B (yellow box, 0–10° E, 15–25° S). All flight days (8, 14, and 24) in September 2016 are marked by asterisk (green vertical lines). The purple shading over the land in (d–f) refers to



**BLH higher than 3250 m. (h) The 2-D joint pdf (shading) with scatter plot are shown with 1-D histogram (pdf in line)**
**of (top) low-CF and LTS and (bottom) low-CF and BLH over region A (left) and B (right) during September 2016.**


As seen in monthly mean low-CF in Figs. 6 and 7, low cloud structure and cover appear to be influenced by the large-scale
subsidence, LTS, and BLH. Previous studies showed that the stratocumulus tends to be positively associated with LTS
(Wood 2015), and LTS can be the best empirical predictor of boundary layer cloud cover (Klein and Hartmann, 1993).
However, the relationship may be weaker when other factors are at play. Here we examine the daily variability of the low-
CF over the course of three flight days in Fig.13. Clearly shown is that LTS and low-CF is positively correlated over SE
Atlantic (Figs. 13(a–c), g). It also appears that deeper BLH tends to reduce the low-CF. For example, on 14 and 24
September 2016, as the LTS is stronger, low-CF increases over most of the regions. Shallow BLH (< 1 km) promotes the
occurrence of a high low-CF as seen on 24 September 2016 (Figs. 13(b, c, f)). However, on 8 September 2016, a midlatitude
weather system has broken up the sheet of low clouds over the south of 20° S, so a reduced low-CF is observed. The LTS is
also weaker on 8 September 2016 compared to other days. (Fig. 11a). It appears that the frontal system mixes out the marine
layer, reducing LTS–low-CF and BLH–low-CF relationship. (Figs. 13(a, d)).

During the September 2016, the LTS –low-CF relationship is even higher further south (region B, R ~ 0.73). The weak
relationships in the flight region near the coast (e.g. 0–10° E, 5–15° S, A) than further west (e.g. 10° W–0, 5–15° S) are
obtained for all deployment months. The joint pdf in Fig. 13h also shows that LTS tends to increase when low-CF increases
while BLH tends to decrease (Fig. 13h) over the flight regions. Some of the "transition" to middle to high clouds is related to
transition to scattered cumulus (see Fig. 6S in the supplementary material, September 24), but note the white blank spots can
be also related to how we define low-CF, rather than some physical process, because the 2.5 km is not physically
constraining definition to some extent.

The key meteorological characteristics during September 2016 deployment can be summarized as follows: 1) AEJ-S
development around 600 hPa is highly associated with heat low (temperature gradient) on diurnal to weekly temporal scales.
2) Dry convection is found over land south of about 10S, with moist convection north of 10° S. The moist plume originated
from the land is advected by AEJ-S and transported offshore at about 3–5 km altitude. There is not much moisture transport
above 500 hPa (~5.5 km).  3) The high low-CF tends to be strongly associated with strong LTS, moderately correlated with
low BLH.  Strong AEJ-S is found at the west of the jet exit region (10° W–0, 5–15° S) with strong LLJ north of 15° S near
the coast (0–10° E, 5–15° S).

**4.2 Deployment year 2 (São Tomè, Aug. 2017)**

The second deployment was performed in São Tomé (0.34° N, 6.73° E). Considering that August is a transition month from
winter (July) to spring (September) in SH, the transition can simply delay. And this actually happened during the August



2017 deployment. Characteristics of synoptic-scale features during the flight days in August 2017 deployment are
590  summarized in Table 2.

**Table 2. Characteristics of synoptic-scale features over SE Atlantic during the August 2017 deployment.**

| Dates | Flight days | Focused Lon/Lat | Synoptic description |
|---|---|---|---|
| 9–12 Aug. | 9, 12 Aug. | 0–20° E, 0–15° S | Slow, unorganized moisture advection by the relatively slow-moving, weak AEJ-S |
| 13–18 Aug. | 13, 15, 17 Aug. | 10° W–20° E, 0–15° S | Strong westerly wind, suppressed AEJ-S, dry condition |
| 18–22 Aug. | 19, 21 Aug. | 30° W–20° E, 5–25° S | Strong moisture advection from land, along with the strengthening of AEJ-S. Strong subsidence over the SE Atlantic Ocean throughout the 600–925 hPa level. Very weak zonal wind on 20–21 August 2017 as strong mid-latitude and upper-level system pushes northward. |
| 22–28 Aug. | 26, 28 Aug. | 20° W–15° E, 5–25° S | Relatively slow-moving moisture advection corresponding to the weakening of AEJ-S. |
| 28–31 Aug. | 30, 31 Aug. | 20° W–20° E, 5–25° S | Relatively fast-moving moisture advection along with enhancement of AEJ-S |

595





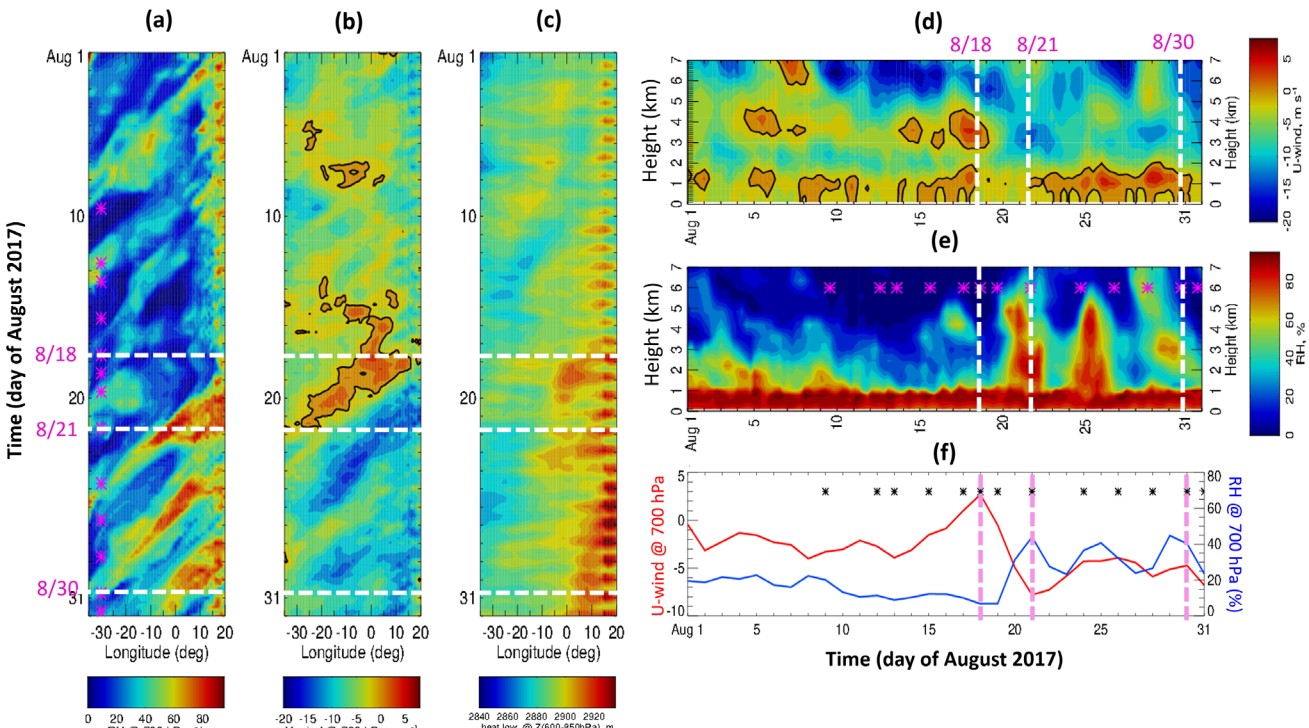

**Figure 14. (a, b) Longitude-time cross-section of: (a) RH at 700 hPa (shading, %), (b) zonal wind at 700 hPa (shading, m s⁻¹), (d–f) altitude-time cross-section at 10° E, all averaged over 5–7° S, and (c) heat low averaged over 10–15° S during August 2017. The black contour in (b) and (d) represents 0 value. The white dashed lines in (d–e) indicate the flight days investigated further in this study, and the asterisk represent the flight days during August 2017 deployment. (f) time series of the zonal-wind at 700 hPa (red line) and RH at 700 hPa (blue line) averaged over 0–10° E and 6–10° S.**

From the mean figures shown earlier (Fig. 3), we found the AEJ-S is centered closer to 5° S in August rather than 10S. Considering that AEJ-S starts to develop in August by strengthening and moving southward, we pick 5–7° S as the latitudinal average region, instead of 8–10° S in August 2017. A Hovmöller diagram shows that a relatively dry condition continues until around 18 August 2017, with prevailing strong westerlies over the Atlantic Ocean. It also shows a weaker heat low than in September 2016 (Fig. 14c). Another interesting feature is that the maximum height of AEJ-S is lower (~ 700hPa, ~3 km) in August than its maximum height in September (Fig. 14d) as shown in Figs. 2 and 5. Most of the aerosol in the boundary layer occurs in August (Zhang and Zuidema, 2019), and this aerosol layer coincides with the lower altitude of the AEJ-S. Relatively dry condition (RH is less than ~60 % ($q \sim 8$ g kg⁻¹)) over the ocean continues until 18 August 2017, and the dry regime shifts to a moist regime after 20 August 2017 when moist plumes develop (Figs. 14(e, f)). The heat low



also gets stronger after 18 August 2017, which coincides with the development of the AEJ-S over the continent. Before the AEJ-S develops, the enhanced TEJ dominates at upper-levels under very dry condition (Figs. 14(d, e)). Note that moisture above the boundary layer and the AEJ-S are at lower altitudes than in September, which is shown in the mean figures in Fig.

615   5.

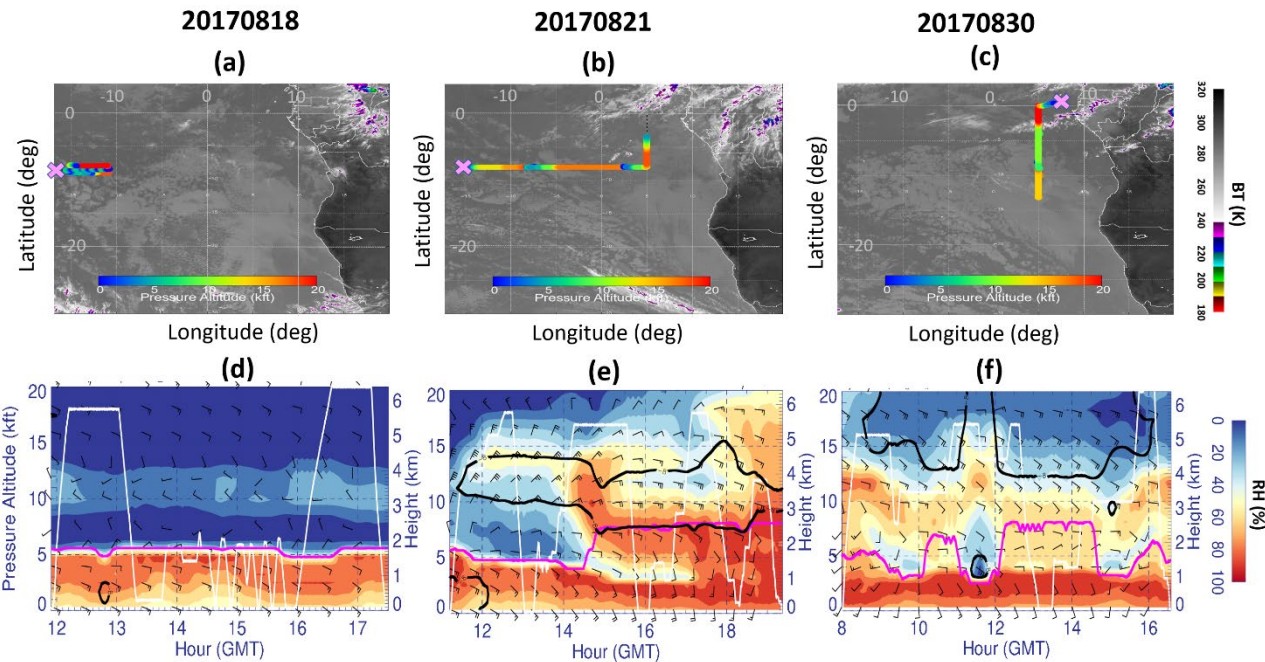

**Figure 15. (a–c) The horizontal flight tracks during August 2017 ORACLES deployment plotted on Meteosat IR 10.8 μm imagery at 1345 UTC. The color represents the altitude of the flight along the horizontal flight track. (d–f) Curtain plot of RH along the flight track during 18, 21, and 30 August 2016. The white contour represents the flight profile. The magenta line in (d–f) represents the BLH along the flight track. Bold black contours in (d–f) are zonal wind (-8 m s⁻¹). The green circle indicates the northeasterly feature during the given flight day (Fig. 15e).**

The flight tracks with altitudes and RH along with the flight during the three August 2017 deployment are shown in Fig. 15. The 18 August 2017 flight was a coordinated flight with CLARIFY out of Ascension island. The 21 August 2017 flight was a transit flight from Ascension to São Tomé. The 30 August 2017 flight was a routine flight to about 13° S. As shown in the

625   Hovmöller diagram in Fig. 14, the dry condition in the free troposphere above the moist marine boundary layer was persistent through 18 August 2017 (Fig. 15d). A thin moist layer in the free troposphere (3~ 4 km, Fig. 14d) is observed on

18 August 2017, but it does not extend very far off the coast. After 18 August 2017, the AEJ-S starts to develop although it is lower than in September.

While 18 August 2017 displays very dry conditions except for the marine boundary layer (~5 kft (1.5 km)), 21 and 30

August 2017 show moist conditions throughout the flight track (Figs. 15(e, f)). Interesting features are the strong moisture tongue developed as the flight approaches the coast on the 21 August 2017 case (Fig. 15e). This high RH "tongue" is tied to the mid-level clouds that moved over the flight track from the northeast, mostly from the north of 5–10° S (Figs. 15(b, e)). The "tongue" structure, however, seems to be also affected by the large-scale northeasterly (18 kft, 6 km around 12–14 hours, marked as circle in Fig. 15e. Contrary to the relatively uniform BLH on 18 August 2017, the BLH on 21 and 30 August 2017

show a lot of the dry and moist layer alternatively intercepting the flight track. The AEJ-S on 21 and 30 August 2017 is strong in these flights, penetrating further west (Figs. 15(e, f)).



**Figure 16.** Map of (a–c) specific humidity (*q*) at 700 hPa (shading, g kg$^{-1}$) and horizontal winds at 700 hPa (vectors, m s$^{-1}$) overlaid by thickness (geopotential height (Z) difference between 600 hPa and 850 hPa) (color contour > 2920 m, 10 m increase; high values over land represents the heat low). The black line represents the horizontal flight track on the given day. Map of (d–f) potential temperature (*θ*) at 850 hPa (shading, K) and horizontal winds at 925 hPa (vectors, m s$^{-1}$) overlaid by Z at 1000 hPa (blue contour, m) at 1200 UTC 18, 21, and 30 August 2017. (g) Time series of daily averaged (top) vertical velocity (ω) at 800 hPa (orange) and low-CF (blue) over region B, and (bottom) AEJ-S



**wind speed (red: horizonal wind at 700 hPa over 0–10° E, 5–15° S, region A), and LLJ wind speed (green: horizontal**

**wind at 925 hPa over 0–10° E, 15–25° S, region B) during August 2017. The asterisk in (g) represents the August 2017**

**flight days. The magenta line refers to the three flight days. (h) 2-D joint pdf of the daily averaged AEJ-S wind speed**

**and LLJ wind speed (correlation is obtained for 18 – 31 August 2017 (blue dots) when the AEJ-S develops). The**

**marginal plot shows their normalized pdf for whole month (gray) and the days during the month when AEJ-S**

**develops (red), respectively.**


After a dry spell in mid-August 2017 until around August 18–19, 2017, the moisture from northern Africa is transported to the ocean by the strong AEJ-S, shown on August 21 and 30, 2017 (Figs. 16(b–c)). The winds at 925 hPa appear to be slightly enhanced on August 21 and 30, especially over the subtropical Atlantic Ocean around 20° W–10° E, 5–20° S (Figs. 16(e–f)). The latitudinal difference in water vapor and $\theta$ between Congo-Zaire basin (5–10° S) and Namibia-Kalahari dryland (15–21°

S) inland gets larger on 21 and 30 August 2017 than 18 August 2017 (Figs. 16(a-f)). The meridional temperature gradient over land around 10° S (Congo-Zaire basin and Namibia-Kalahari dryland) is evident for all selected flight days, shown in Figs. 16(d-f).

The strong AEJ-S and the recirculating wind associated with the jet on 21 August 2017 extend to the ocean, and they mainly interact with the mid-latitude jet (Figs. 16(b, e)). It is unclear why the development of AEJ-S is suppressed on 18

August 2017, even though offshore/inland and Congo-Zaire basin/Namibia-Kalahari dryland temperature gradient is relatively strong (Fig.16g). One explanation for this can be 1) a change in low-level circulation (925–850 hPa), and 2) a weak heat low positioned off from the continent on August 18 (Fig. 16a). AEJ-S is also correlated with LLJ in August like in September (Fig. 11h), but AEJ-S–LLJ correlation only becomes significant after AEJ-S starts to develop after 18 August 2017.

The developing subtropical high-pressure system (Figs. 16(d–f)) is clearly shown for 21 and 30 August 2017, but this is barely observed on 18 August 2017. The high low-CF is also weakly associated with reduced subsidence in August, and this relationship gets higher toward the end of August (upper panel of Fig. 16g). However, the subsidence–low-CF relationship is not very clear on a daily time scale, but more meaningful on monthly mean time scales (De Szoeke et al., 2016; Adebiyi et al. 2018). Subsidence in August 2017 is slightly weaker than in September 2016.




**Figure 17.** (a–c) *Ocean*: latitudinal cross-section of RH (shading, %), horizontal winds (wind barbs, green, m s⁻¹), θ (navy, K), and BLH (magenta, m) at 0° E (top) and skew-T log-P diagram averaged over 0° E and 9–10° S (bottom: from the left) at 1200 UTC 18, 21, and 30 August 2017. Bold black contours are zonal wind (-8 m s⁻¹). (d–f) *Land*: the



**same as in (a–c) except for the cross-section at 18° E and Skew-T log-P diagram averaged over 18° E and 5–6° S (CZ: Congo-Zaire basin) and 18° E and 16–17° S (NK: Namibian-Kalahari dryland). The gray filled area represents the topography. The vertical lines in the cross-section plots refer to the latitude we examine.**


High RH (> 60 %) is confined over the continents, limiting the moist plume transport on August 18, 2017 (Fig. 17a). AEJ-S develops after 18 August 2017, advecting moisture from the continent at 700 hPa (Fig. 17b) offshore. Enhanced AEJ-S features are shown on August 21 and 30. Temperature inversion is also observed over the ocean, and the marine boundary layer is consistently moist over the ocean, forming stratocumulus. Note that the moist and dry convection over land is much more effective at moistening the mid-levels (700–600 hPa) on 21 and 30 August 2017 than on 18 August 2017.

The cross-sections suggest that the horizontal temperature gradient is weak, which makes the AEJ-S weaker (Fig. 17a). Note that the moisture at all levels does not extend as far south as in September. Over the land, the temperature inversion layer sits above the unsaturated, dry layer over the land, and dew point temperature follows the constant water vapor mixing ratio (21, 30 August  in the bottom panels of Figs. 17(e, f)), confirming dry convection occurs after the dry spell around 20 August 2017. Considering dry condition before 20 August, convection is less likely initiated during this dry spell over the dryland in August 2017. Over the dryland (NK: 18° E, 16–17° S), temperature increases at the surface from 18 August to 30 August, and the dry convection layer deepens, consistent with the stronger heat low as the month progresses (Figs. 17(d-f)). This results in more moisture south of 10° S in the 850–600 hPa region later in the month. Over the congo basin (CZ: 18° E, 5–6° S), the depth of the relatively moist layer a bit deepens on 21 August and becomes shallow on 30 August 2017. Unlike September 2016, CAPE is generally low. This is consistent with the lower rainfall in that month as compared to September.


**Figure 18.** Map of (a–c) low-CF (shading, %) overlaid by LTS (contour, K) and horizontal winds at 925 hPa (black vectors, m s⁻¹), (d–f) BLH (shading) overlaid by horizontal winds at 925 hPa (vectors, black), and (g) (top) daily time series of low-CF (blue lines, %) and LTS (red lines, K), and (bottom) low-CF (blue line, %) and BLH (magenta lines, meter) during August 2017. The solid boxes in (a) are averaged over region A (magenta box, 0–10° E, 5–15° S) and dashed lines are averaged over region B (yellow box, 0–10° E, 15–25° S). All flight days (18, 21, and 30) in August 2017 are marked by asterisk (green vertical lines). The purple shading over land in (d–f) refers to BLH higher than 3250 m. (h) The 2-D joint pdf (shading) with scatter plot are shown with 1-D histogram (pdf in line) of (top) low-CF and LTS and (bottom) low-CF and BLH over region A (left) and B (right) during August 2017.



During August 2017, the low-CF is positively correlated with LTS, as in September 2016 (R~ 0.57, top panel of Fig. 18g).
The BLH tends to be high over the open ocean (10–40° W, 10–40° S) and land, and tends to be shallow near the west coast
of Africa (Figs. 18 (d–f)). The high low-CF is weakly associated with the low BLH (e.g. 22 August 2017 in Fig. 18g).The
pdfs of LTS, BLH, and low-CF shows that they have different distribution over these regions (A and B) (Fig. 18h), although
this single pdf does not explain the spatiotemporal correlation of LTS and BLH with low-CF. The pdf of LTS shows that
LTS is generally lower in August than September over these regions, implying that the atmosphere is less stable in August
(compare with Fig. 13h). Possibly, this is because 1) surface warming due to warmer SST in August than in September over
SE Atlantic, 2) low-level atmospheric cooling above the top of low-cloud, and 3) strong mid-latitude frontal system
developed particularly in August 2017. It will be particularly interesting to see how aerosol can influence the atmospheric
cooling by interacting with low-cloud, but this is beyond scope of this study. The low CF is positively correlated with LTS;
and shows the strongest correlation in Region B. We hypothesize that there is greater variability in both low CF and LTS in
region B because of the recurrent cloud clearing associated with midlatitude frontal systems.

To summarize, the key meteorological characteristics during August 2017 deployment are as follows: 1) AEJ-S develops
around 700 hPa (~ 3 km, slightly lower than those in September and October), and this is associated with heat low (driven by
latitudinal temperature gradient), but both of them are not as strong as those in September (note that AEJ-S also moves south
during the month). 2) The enhanced low-CF tends to be associated with high LTS, shallow to moderate BLH (600–1200 m),
3) After 20 August 2017, a layer of dry convection to 700 hPa is found over the Namibian-Kalahari dryland, and a moist
plume originated from the land is advected by AEJ-S and upper-level jet at about 5–9 km altitude.

### 4.3 Deployment year 3 (São Tomè, Oct. 2018)

The final deployment of ORACLES was performed in October 2018. October is also a month of transitioning from cool to
warm season in southwestern Africa, pushing precipitation further south (Fig. 1), and this reduces the BB substantially. The
low-level and mid-level wind further changes its directions, with moisture transport modified by the changes in the west
African monsoonal circulation. The AEJ-S is still prevailing, but temperature gradient over the land is weak and the
variability is large due to 1) the convective system migrating from the north of 5° S associated with the seasonal change and
2) the mid-latitude frontal system. The characteristics of synoptic-scale and convective features during the flight days in
October 2018 deployment are summarized in Table 3.

**Table 3. Characteristics of synoptic-scale features over SE Atlantic during the October 2018 deployment.**

| Dates | Flight days | Focused Lon/Lat | Synoptic description |
|---|---|---|---|
| 1–6 Oct. | 2, 3, 5 Oct. | 20° W–20° E, 0–25° S | Fast-moving (~12.8 m s$^{-1}$) convection mostly from developing continental |





| | | | convection (< 5° N) marching westward along with AEJ-S. |
|---|---|---|---|
| 6–10 Oct. | 7 Oct. | 20° W–20° E, 0–25° S | Moderate convection development. |
| 11–12 Oct. | 12 Oct. | | AEJ-S weakening, weak moisture advection from the African continent. |
| 14–22 Oct. | 15, 17, 19, 21 Oct. | 30° W–10° E, 5–25° S | Slowly propagating (~6.4 m s$^{-1}$) moisture transport from the land to the ocean. Further moisture transport as AEJ-S gets stronger. Subsidence and large-scale anticyclone are enhanced over the SE Atlantic ocean. Mid-latitude and upper-level flows are tied to the lower atmospheric circulation. |
| 22 –28 Oct. | 23, 25 Oct. | 20° W–20° E, 5–25° S | Convection marches further south (< 20° S). Cooling down of temperature and a weakening of both latitudinal moisture and temperature gradient over land. A weakening of AEJ-S. |
| 28–31 Oct. | | 20° W–20° E, 5–25° S | Moisture transport occurs from land to ocean, along with the restrengthening of AEJ-S. |



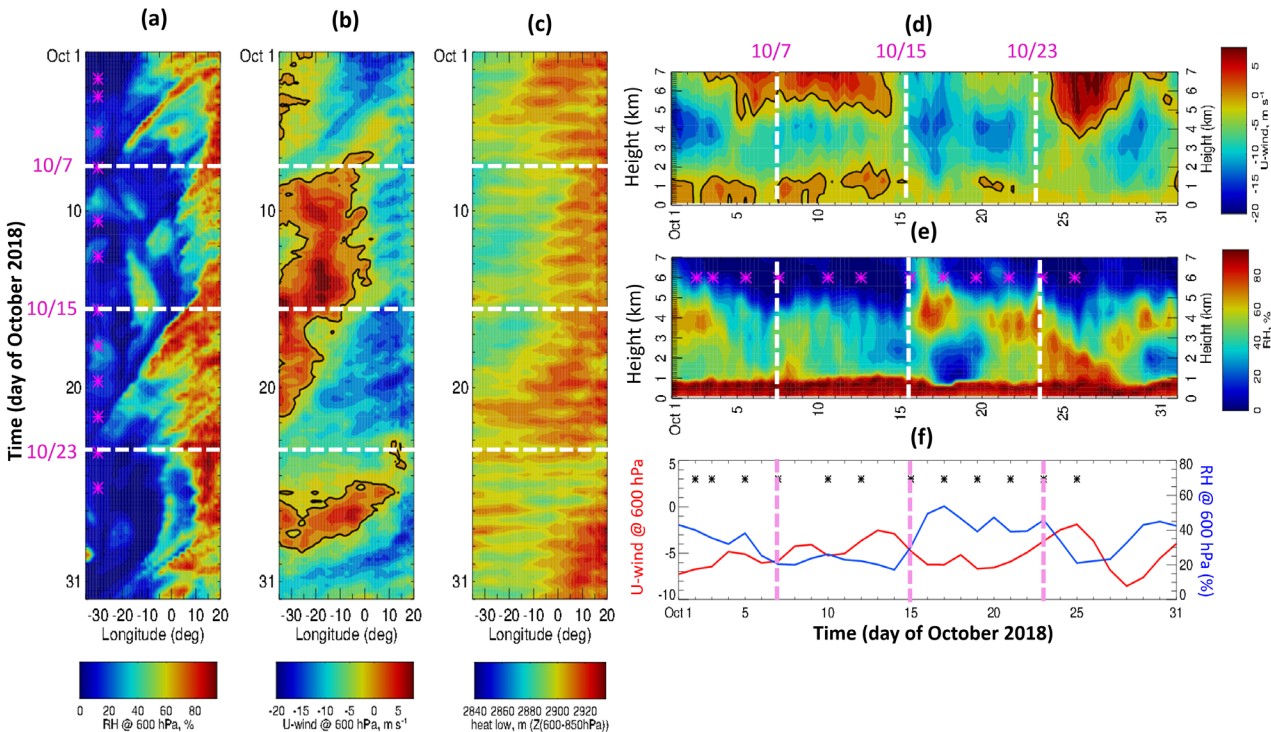

**Figure 19. Longitude-time cross-section of: (a) RH at 600hPa (shading, %), (b) zonal wind at 600hPa (shading, m s⁻¹),** (c) heat low, and (d-e) altitude-time cross-section at 10° E, all averaged over 8–10° S during October 2018. The black contour in (b) and (d) represents 0 value. The white dashed lines indicate the flight days investigated further in this study, and the asterisks represent the flight days during October 2018 deployment. (f) Time series of the zonal-wind at 600 hPa (red line) and RH at 600hPa (blue line) averaged over 0–10° E and 6–10° S.

Figure 19 shows the Hovmöller diagrams of RH and zonal-wind at 600 hPa and zonal-, meridional-wind at 925 hPa, and BLH plots during October 2018 deployment. As shown in Fig. 19a, variability ranging from diurnal to weekly time scale is observed in RH and wind. The modulation of AEJ-S by heat low is clearly shown as in other deployments (Figs. (9, 14)). AEJ-S leads high RH by 1–2 days, indicating that AEJ-S can transport elevated moisture from land to the ocean (Figs. 19(a, b), (d, e)).

The flow patterns look quite different between the distinct time periods. For example, during 15–22 October 2018, high RH propagates over the ocean due to the strong AEJ-S. Towards the end of October around 23–28 October 2018, high RH plume predominates over the continent, without westward propagation, which is attributed to the weakened AEJ-S (Figs. 19(e, f)). During this period, the mid-level wind is weak easterly, while low-level winds are also very weak southeasterly (Fig. 19d).





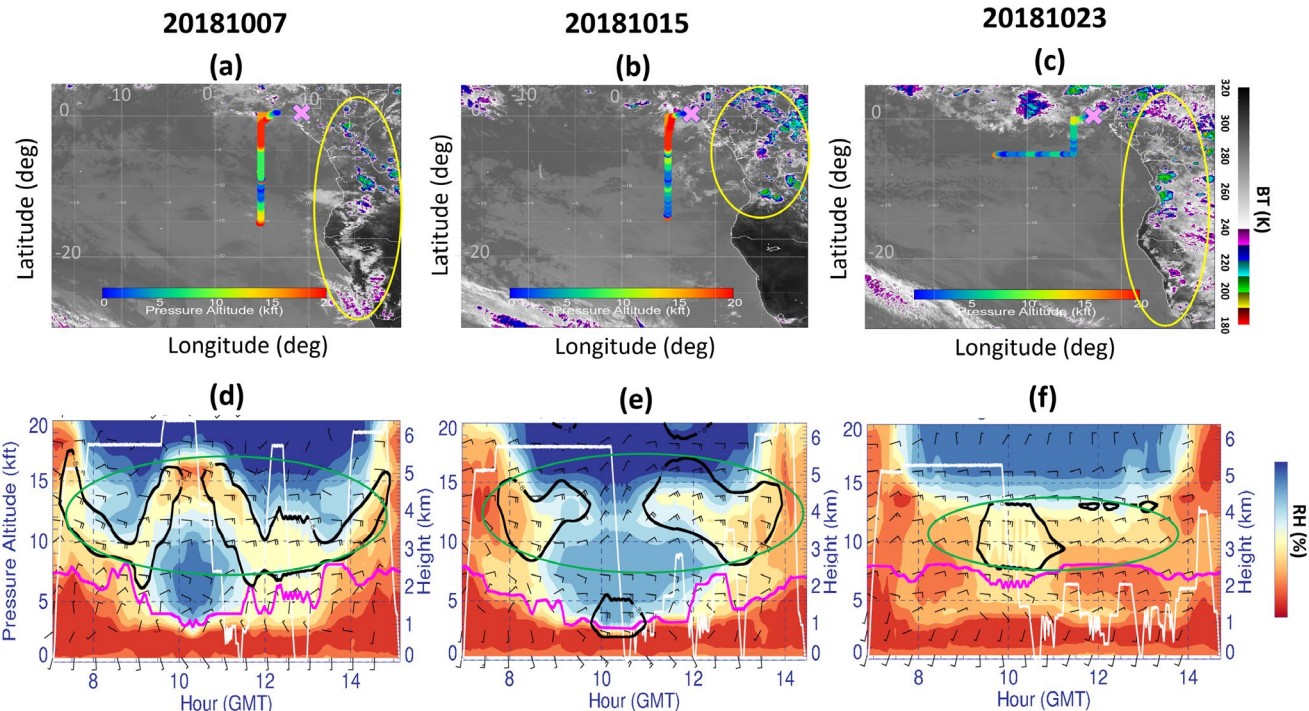

**Figure 20. (a–c) The horizontal flight tracks during October 2018 ORACLES deployment plotted on Meteosat IR 10.8 μm imagery at 1345 UTC. The color represents the altitude of the flight along the horizontal flight track. (d–f) Curtain plot of RH along the flight track during 7, 15, and 23 October 2018. The white contour represents the flight profile. The magenta line in (d–f) represents the BLH along the flight track. Bold black contours in (d–f) are zonal wind (-8 m s⁻¹). The yellow circles refer to the incidence of deep convection over the African continents (Figs. 20(a–c)), while the green circle indicates the moisture transport features during the given flight days (Figs. 20(d–f)).**

Satellite imagery during the October deployment is shown in Fig. 20. Substantial deep convection develops over the continent on most days in October 2018, with low brightness temperature (< 230 K), indicating deep convection in the north of 10° S marching down to the south (Figs. 20(a, c)). Along the flight tracks, high RH is observed (RH > 60%) for all three cases, with the highest RH during the low latitude leg on 23 October 2018 (Fig. 20f). During the middle of the 7 October 2018 flight, strong mid-level easterly and southeasterly winds carry high RH plume, leading to relatively moist condition above dry atmosphere (~ 15 kft, 4.5 km), which is associated with moisture transport by the mid-level AEJ-S (Figs. 20(a, d)), which develops further south than 8–10° S.

A similar feature is found on 15 October 2018 case, but less moist air is found on the top of dry air, which may be associated with the northward displacement of RH on that day. Basically, the AEJ-S is strengthening, and the RH is starting to propagate westward (Figs. 20(b, e)). The aircraft reached the southern edge of the AEJ-S on that day (Fig. 21b). High RH





condition is observed throughout the entire flight track on 23 October 2018 (Fig. 20c). Much moist air is shown along the flight track, along with the elevated marine BLH on that day (Fig. 20f). The weakening of the easterly wind, indicative of AEJ-S (~ 15 kft, 4.5 km), is also observed (Fig. 20f). That is because the AEJ-S is displaced northward in 23 October 2018

775 (See Fig. 21c). This explains why the AEJ-S appears much weaker in the Hovmöller averaged over 8–10° S. The horizontal thickness gradient (heat low) over the continent is also much weaker on 23 October 2018 (Fig. 21f) as compared to other two cases (Figs. 21(d, e)). We speculate that this may be due to the extensive cloud cover limiting the sensible heating over the Kalahari-Namib dryland.



**Figure 21.** Map of (a–c) specific humidity ($q$) at 600 hPa (shading, g kg$^{-1}$) and horizontal winds at 600 hPa (vectors, m s$^{-1}$) overlaid by thickness (geopotential height ($Z$) difference between 600 hPa and 850 hPa) (color contour >2920 m, 10 m increase; high values over land represents the heat low). The black line represents the horizontal flight track on the given day. Map of (d–f) potential temperature ($\theta$) at 850 hPa (shading, K) and horizontal winds at 925 hPa (vectors, m s$^{-1}$) overlaid by $Z$ at 1000 hPa (blue contour, m) at 1200 UTC 7, 15, and 23 October 2018. (g) Time series of daily averaged (top) vertical velocity ($\omega$) at 800 hPa (orange) and low-CF (blue) over region B, and (bottom) AEJ-S wind speed (red: horizonal wind at 700 hPa over 0–10° E, 5–15° S, region A), and LLJ wind speed (green: horizontal wind at 925 hPa over 0–10° E, 15–25° S, region B) during October 2018. The asterisk in (g) represents the October 2018 flight days. The magenta line refers to the three flight days. (h) (h) 2-D joint pdf of the daily averaged AEJ-S wind speed and LLJ wind speed (correlation is obtained for 1–24 October 2018 (blue dots) when the AEJ-S develops). The marginal plot shows their normalized pdf for whole month (gray) and the days during the month when AEJ-S develops (red), respectively.

As convection becomes more active during the October deployment, the continental heating weakens and the latitudinal $\theta$ gradient decreases. While the AEJ-S is stronger on 7 and 15 October 2018, it is much weaker on 23 October 2018, which can be tied to the reduction in the meridional gradient of RH and $\theta$ at 10° S (Figs. 21(a–f), Fig. 22). The cool temperature over the continent on 23 October 2018 is associated with the southward movement of moisture and convection from the north (Fig. 21c)). The westward extent of AEJ-S tends to be associated with the shape and strength of the continental moisture transport offshore (Figs. 21(a–c)), and this is also tied to the latitudinal temperature gradient (Figs. 21(d–f)). Basically, the dome of hot dry air over the southern African highlands has disappeared by 23 October 2018, consistent with a breakdown of the 600 hPa anticyclone over southern Africa (the cross section in Fig. 22). The anticyclone reforms toward the end of the month. Possible causes are: 1) mid-latitude waves associated with strong upper-level disturbance influence the break down of the dome of hot dry air (Fig. 21f), 2) convection reduces the temperature gradient by increasing high cloud cover over the region, reducing surface heating (Figs. 20c, 21c).

As in the September and August deployment, the AEJ-S during October 2018 is also associated with the LLJ with a lag: the AEJ leads the LLJ about 0–2 days (Figs. 21(g, h)). The AEJ-S and LLJ are both strong during October 2018 and strong LLJ tends to be tied to reduced subsidence and reduced low-CF during October (Figs. 21 (g, h)). Of the three deployment months, the AEJ-S–LLJ relationship is the strongest in October. Reduced subsidence, a sufficiently strong temperature inversion, moist and warm condition associated with mid-latitude frontal system may provide favorable conditions for both AEJ-S and LLJ to develop together, but the clear mechanism and how they affect low-CF needs to be further investigated.

In October 2018, the enhanced large-scale subsidence tends to increase low-CF off the Namibian coast (Fig. 21g). In general, the similar characteristics are also found in the mean field (Fig. 6), although this association is highly variable depending on regions and time scale. In fact, these different features of the subsidence–low-CF relationship shown in deployment months can be interpreted as the result of the interaction between subsidence, temperature inversion, and





cloudiness (Klein and Hartmann 1993; Muñoz et al. 2011), emphasizing that the primary mechanism for modulating cloud
cover is temperature inversion (Myers and Norris, 2013). Furthermore, subsidence occurrence at different time scales and its
impact on low-CF is also more significant at monthly time scale than daily to synoptic (De Szoeke et al., 2016; Adebiyi and
Zuidema, 2018). Since temperature inversion can be associated with atmospheric stability (e.g. LTS), the large day-to-day
variability in the stratocumulus layer can be also partially associated with the passage of frontal systems from the
midlatitudes.








**Figure 22. (a–c)** *Ocean*: latitudinal cross-section of RH (shading, %), horizontal winds (wind barbs, green, m s⁻¹), potential temperature (θ) (navy, K), and BLH (magenta, m) at 0° E (top) and Skew-T log-P diagram averaged over 0° E and 9–10° S (bottom: from the left) at 1200 UTC 7, 15, and 23 October 2018. Bold black contours are zonal wind (-



**8 m s⁻¹). (d–f) *Land*: the same as in (a–c) except for the cross-section at 18° E and skew-T log-P diagram averaged over 18° E and 5–6° S (CZ: Congo-Zaire basin) and 18° E and 16–17° S (NK: Namibian-Kalahari dryland). The gray filled area represents the topography. The vertical lines in the cross-section plots refer to the latitude we examine.**

The latitudinal cross-section of RH along with the horizontal wind shows that the transport of moist plumes to the ocean is accompanied by AEJ-S in October, which is also shown in the other months (top panels of Figs. 22 (a–c)). Over the ocean,

the sounding suggests that low-cloud layers may form around 925 hPa, as we also see over the ocean in August and September flight days. (Figs. 12, 17, 22 (a–c)). All flight days discussed here show temperature inversion layers, developed over the saturated air (i.e., the temperature line nearly meets the dew point temperature line). This indicates the presence of stratocumulus over the ocean (Figs. 22 (a–c)**).**

Over the land, dry convection is still dominant on 7 and 15 October 2018 over the Namibia-Kalahari dryland (18° E,

16–17° S). However, over the Congo-Zaire basin (18° E, 5–6° S), the dry convection disappears as dew point temperature no longer follows the constant water vapor mixing ratio line and gets closer to temperature, transitioning to deep moist convection (top panel of sounding in Figs. 22(d–f)). On 23 October, the moist plumes with high RH (> 60%) reach 10–20° S (Figs. 22(d–f)) with moist convection over both Congo-Zaire basin and Namibia-Kalahari dryland (see satellite image, Fig. 20c). Deep cloud layers can be identified by locations where the dew point temperature and the environmental temperature curves get very close for deep vertical layers, which is clearly shown on 23 October 2018 over land (soundings in both CZ

and NK in Fig. 22f). In this case, deep convection presents over the continent with almost the entire troposphere up to 200 hPa saturated (sounding plots in Fig. 22f). With such deep clouds, RH reaches ~ 100 %, indicative of heavy rainfall. Satellite imagery shows deep convection developing over southern Angola and Zambia starting on 19 October 2018, spreading southward to northern Namibia and Botswana by 23 October 2018. The temperature gradient over the land disappears (as

shown by Fig. 22f), and so does the AEJ-S over land and near coastal region (Fig. 21c).



**Figure 23.** Map of (a–c) low-CF (shading, %) overlaid by LTS (contour, K) and horizontal winds at 925 hPa (black vectors, m s$^{-1}$), (d–f) BLH (shading) overlaid by horizontal winds at 925 hPa (black vectors, m s$^{-1}$), and (g) (top) daily time series of low-CF (blue lines, %) and LTS (red lines, K), and (bottom) low-CF (blue line, %) and BLH (magenta lines, meter) during October 2018. The solid lines are averaged over region A (magenta box, 0–10° E, 5–15° S) in (a) and dashed lines are averaged over region B (yellow box, 0–10° E, 15–25° S). All flight days (7, 15, and 23) in October 2018 are marked by asterisk (green vertical lines). The purple shading over the land in (d–f) refers to BLH higher





**than 3250 m. (h) The 2-D joint pdf (shading) with scatter plot are shown with 1-D histogram (pdf in line) of (top) low-**
**CF and LTS and (bottom) low-CF and BLH over region A (left) and B (right) during October 2018.**

The positive relationship between LTS and low-CF is also shown in October (e.g. 10° W–0, 5–15° S), but the correlation is much weaker and statistically insignificant than in August and September (see Fig. 24). In October, when the subsidence weakens compared to August and September, low-CF and LTS are not linearly correlated but they are still associated,
especially at intermediate values (16 – 20 K) of LTS (Fig. 23h, Zhang et al., 2009). Like August and September, LTS–low– CF correlation in October is more evident in region B than in region A (Figs. 13, 18h), although the correlation strengths get small. As the LTS gets stronger, the low–level CF gets larger, as shown in Figs. 23(a–c). BLH tends to be high over the ocean (10–40° W, 10–40° S) and tends to be shallow near the west coast of Africa (Figs. 23 (d–f)). In particular, the low-CF decreases on 23 October 2018  (Figs. 23(c, g)), but the high cloud cover increases (not shown), indicating not only day-to-
day variability but also a cloud regime "transition" may affect the cloud cover during this day. The weak inversion with warmer SST condition like October 2018 can favor decoupling (separation) of boundary layer, leading to development of the cumulus layer (Wyant et al., 1997). The low clouds also tend to respond to the change in the BLH; high low-CF is weakly associated with the low BLH (e.g. 6, 18, 25, and 30 October 2018; the bottom plot in Fig. 23g), but the overall correlation between BLH and low-CF over October 2018 is insignificant (Fig. 23g). This can be partially explained by large day-to-day
variability during October.

In short, the meteorological characteristics during the October 2018 deployment can be summarized as follows: 1) AEJ-S develops around 600 hPa (~ 4 km), driven by the continental heat low. 2) However, AEJ-S diminishes as continental convection and precipitation march southward. 3) The relationship among low-CF, BLH, AEJ-S, and LLJ are not statistically significant in most regions of interest. However, i) LTS is positively associated with low-CF over the mid SE Atlantic (10°
W–0, 5–15° S), ii) LLJ is negatively correlated with low-CF south of 15S (10° W–10° E, 15–25° S). 4) Strong temperature inversion is observed over the ocean while no inland temperature inversion is found over both Congo-Zaire basin and Namib-Kalahari dryland.


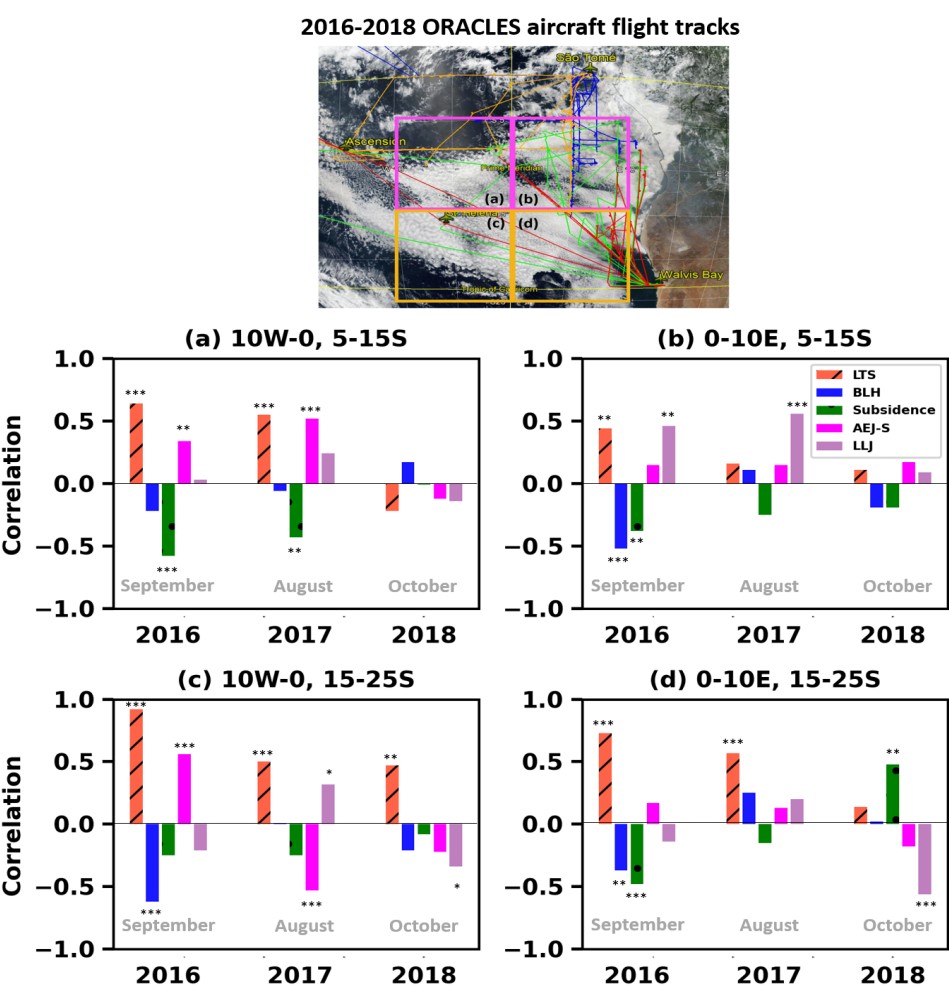

**Figure 24. Bar plots of the linear correlation between the low-CF and the meteorological variables (LTS, BLH, subsidence, AEJ-S, and LLJ) averaged over four subregions in the ORACLES flight regions over SE Atlantic Ocean ((a) 10° W–0, 5–15° S, (b) 0–10° E, 5–15° S (c) 10° W–0, 15–25° S, and (d) 0–10° E, 15–25° S) during the deployment months (marked in the boxed regions over top image with flight tracks). *** (**, \*) denotes when the correlation is statistically significant within 99% (95, 90 %) confidence interval (*p-value* < 0.01 (<0.05, < 0.1)). The top panel represents the ER-2 flight tracks in 2016 (green), and P-3 flight tracks in 2016, 2017, and 2018 (red, orange, and blue, respectively). The top panel with flight tracks on a MODIS image and © Google Maps on 13 September 2018 is adapted from Redemann et al. (2021).**

The linear relationship of meteorological variables to low-CF over SE Atlantic regions during each deployment month at a daily time scale is shown in Fig. 24. Low-CF is positively associated with LTS for all deployment months over SE Atlantic



regions except for the north of 15° S (10° W–0, 15–5° S, Fig. 24c) in October. BLH is also negatively associated with low-CF, especially in September. For all variables, the relationships to low-CF are most pronounced in September for all subregions and show large variability in August and October. Subsidence is negatively associated with low-CF except for south of 15° S near the coast in October (Fig. 24d) at a daily time scale. AEJ-S and LLJ is positively correlated to low-CF in September and August north of 15° S, but it is negatively correlated in October (Fig. 24d). Subsidence strength influence AEJ-S, so that this may be related to this low-CF relationship, as shown in the jets (AEJ-S, LLJ)–subsidence relationship to low-CF is reversed both north and south of 15° S near the coast (0–10° E, Figs. 24 (b, d)) except for September south of 15° S. Over the region far off the coast (10° W–0), the LLJ–low-CF relationship gets weaker but the AEJ-S–low-CF relationship gets stronger than those near the coast.

Importantly here, both the LTS –low-CF relationship is lowest north of 15° S near the coast, in the vicinity of AEJ-S, moisture, and BB aerosol transport from the continent (Fig. 24b). The AEJ-S–low-CF relationship is also diminished near the coast (Figs. 24(b, d)). This can be consistent with finding that the LTS–low-CF relationship can be weaker due to aerosols through a reduction of LTS (Adebiyi and Zuidema, 2018). The multi-year analysis will be needed to quantify the influence of each meteorological variable on low-CF at different time scales.

## 5. Summary and discussion

This paper describes the meteorological factors controlling aerosol transport and low cloud during the August, September, and October 2016–2018 deployments of ObsErvation of Aerosols above CLouds and their intEractionS (ORACLES) project.

### A) Meteorological factors that affect aerosol transport:

- The direct aerosol transport across the African coast to the Southeastern (SE) Atlantic can be greatly influenced by the southern African easterly jet (AEJ-S), represented by the zonal wind at 600–700 hPa < -6 m s$^{-1}$) with entrance over land, and exit over the ocean around 5–10° S, 0–10° E. The AEJ-S is driven by a heat low over the African continent south of 10° S, due to large horizontal temperature gradient between cool, wet air over Congo-Zaire basin (5–10° S) and warm, dry air over Namibia-Kalahari dryland(15–21° S). The height of maximum core of AEJ-S in August (~3 km) is lower than those in September and October (~4 km). *AEJ-S is also associated with the rising motion over land through the secondary circulation, which can impact aerosol lifting, transport, and stratocumulus decks.*

- Moist plumes originating from the African continent are advected offshore by the AEJ-S. The AEJ-S may affect aerosol transport indirectly via its impact on moist plume transport over the ocean with a 1–2 day lag. In August and September deployments, dry convection is dominant over land, especially over Namibian-Kalahari dryland. The moist plume originated from the north (5–10° N) over the continent is advected to the ocean by AEJ-S. Moist





convection creeping down from the north of 10N develops over land at the end of October 2018. *Thus, this westward transport of moist plumes from dry and moist convection by AEJ-S can indirectly affect aerosol transport.*

- Benguela low-level jet (LLJ) develops along the Namibian coast of the southeast Atlantic around 0–10° E, 15–25° S at 925–950 hPa (< 800 m) and at the eastern edge of the prevailing St Helena High and the stratocumulus deck (Nicholson, 2010). It is positively linked to the AEJ-S, as AEJ-S leads LLJ about 0–2 days, especially in October indicating the association with mid-latitude frontal systems. The AEJ-S appears to be also closely tied to the large-scale subsidence over the SE Atlantic off the Namibian coast; weak AEJ-S supports strong subsidence while strong AEJ-S is related to weak subsidence off the Namibian coast. *AEJ-S together with LLJ and large-scale subsidence can also affect the aerosol transport.*

**B) Meteorological factors affecting low-level cloud:**

- The low-CF was enhanced when low-level tropospheric stability (LTS) was high and boundary layer height (BLH) was relatively shallow and moderate (<~1500 m). The correlation between low-CF and LTS is largest in September. The correlation between BLH and low-CF breaks down when large-scale frontal systems pass by such as in October 2018. Temperature inversion layers were observed near the surface during all deployment months over the ocean, indicating the formation of stratocumulus. The correlation between low-CF and LTS decreases near the coast in SE Atlantic, and these reduced relationships are observed during all deployment months. *The low- CF is strongly correlated with LTS and moderately correlated with BLH during the deployment months, and the correlation is highest in September 2016.*

- Strong large-scale subsidence off the coast of Namibia and SE Atlantic ocean tends to be associated with the reduced low-CF in August and September, and enhanced low-CF in October. Enhanced anticyclones associated with the subtropical high-pressure system over SE Atlantic is also tied to low-CF. *The subsidence-low-CF relationship seems to be stronger at the monthly mean time scale than the daily time scale and there is sign change from August and September to October.*

- A robust relationship between AEJ-S, LLJ, and low-CF during the deployment is not obtained due to large spatial and temporal variability. However, the relationship of AEJ-S–LLJ to low-CF appears to be closely linked to subsidence and high-pressure system over SE Atlantic, and strong LLJ tends to reduce low-CF, particularly in October. The day-to-day variability of low-CF at the end of August and October is also largely affected by the mid-latitude frontal system over the South Atlantic Ocean. *The relationship between AEJ-S, LLJ, and low-CF varies among the deployment months, and seems to be affected by the mid-latitude frontal systems over SE Atlantic.*

- The low-CF–sea surface temperatures (SST) relationship is weak or statistically insignificant over SE Atlantic, particularly at the time scale we examine for each deployment month. The SST is slightly higher (~0.5 K) than the





climatological mean during all deployment months. However, SST shows large spatial variability, and its effect
seems to be larger over the north of 10° S and the west of 5° W rather than the deployment regions. In general,
strong LLJ is associated with cool SSTs and a decrease in low-CF in October 2018, especially at the south of 15° S
(15–25° S). *The low-CF–SST relationship is weak or statistically insignificant over SE Atlantic during the
deployment months, particularly at the daily to synoptic time scale.*

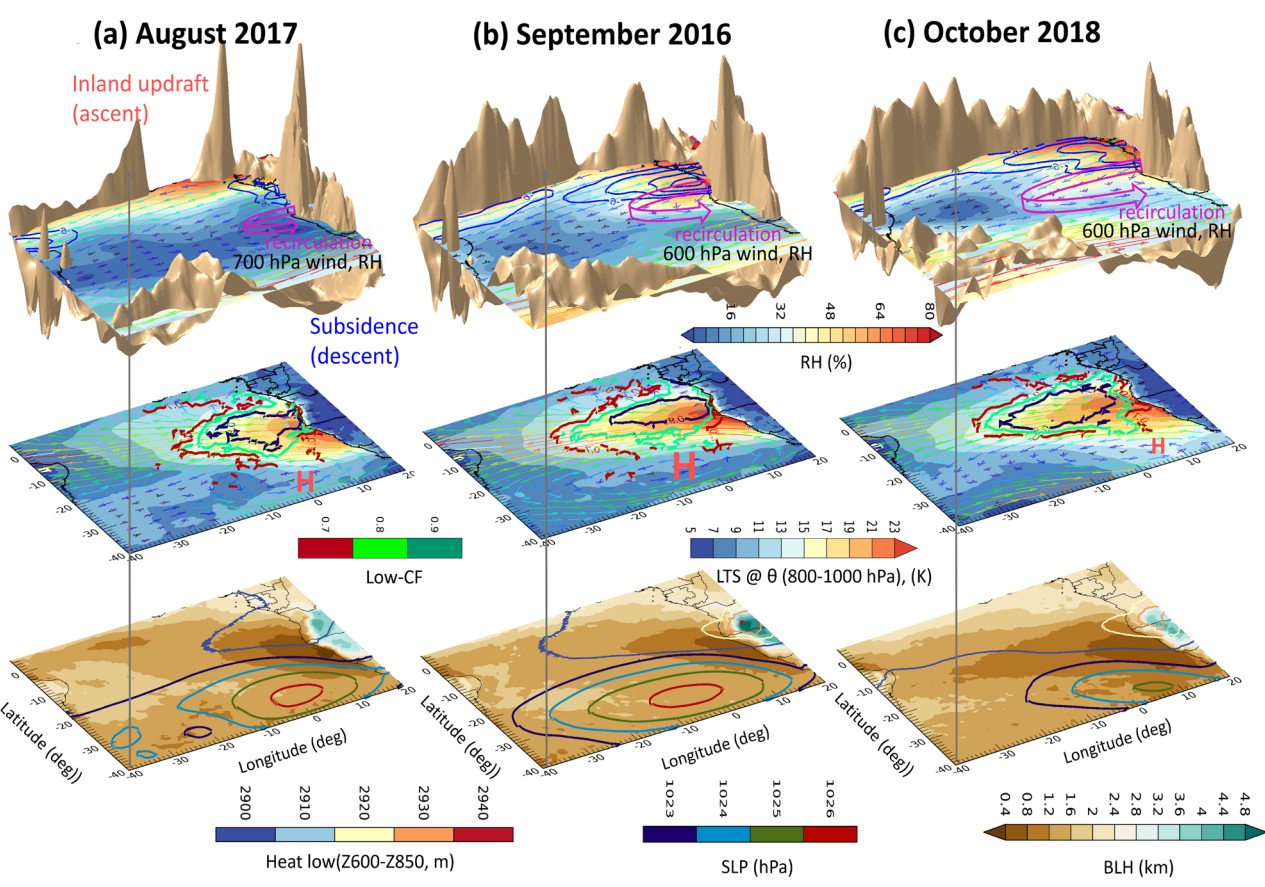


**Figure 25. The visualization of (top panels) map of RH (color, %) and horizontal wind at 700 hPa (vector, m s⁻¹) for
August 2017 and at 600 hPa for September 2016 and October 2018, and vertical motion at 800hPa (gold bumpy
surface, ω multiplied by -1 (i.e. - ω), hPa day⁻¹; positive value to represent the ascent). Above the RH contour plane is
updraft and below the RH contour plane is subsidence. (Middle panels) map of LTS (color, K) overlaid by the**
**horizontal wind at 925 hPa (vector, m s⁻¹) and low-CF (color contours, 0.7–0.9 %). (Bottom panels) map of BLH
(color shading, km) overlaid by thickness between 600 hPa and 850hPa (color contours, meter) and SLP (color**





**contours, hPa) during (a) August 2017, (b) September 2016, and (c) October 2018. The pink arrow in top panels represents the recirculating flow to the African continent. The red "H" mark in mid panels refers to the anticyclones associated with St. Helena High.**

**C) key meteorological characteristics during the deployment months:** The key characteristics during each deployment month is shown in Fig. 25.

- *August 2017:* The maximum AEJ-S occurred at a lower altitude (~ 3 km, ~700 hPa) and further north (5–7° S) compared to the jet in September and October, generating the mid-level anticyclones associated with AEJ-S. The AEJ-S in August transported the moist plume from land to ocean around 0–15° S (top panel of Fig. 25a). AEJ-S–

LLJ relationship was statistically significant when AEJ-S developed and the correlation is higher in the south of SE Atlantic (10° W–10° E, 15–25° S). The heat low was weaker than the other deployment months (September, October). The subsidence was strong over the Southwestern Atlantic Ocean (20–40° W, 30° S) and off the Benguela coast, leading to suppressed AEJ-S. The St. Helena high was positioned near the Benguela coast (Fig. 25a), and a pretty strong heat low was also found throughout the month. Dry convection over Namibia-Kalahari

dryland was found at the end of August 2017 (particular after around August 20). Low-CF was positively correlated with LTS.

The AEJ-S was slightly weaker in August 2017 compared to the climatological mean, but the difference was small (~ 1 m s$^{-1}$). Drier condition was found (region around 5–10° S, 0–15° E) than climatology while moister and wetter condition with high RH mostly resides over northern Africa (< 5N) than the climatological mean. The LLJ

was stronger in August 2017 (~ 2 m s$^{-1}$) compared to the climatological mean. The BLH in August 2017 is lower (~ 100 m) near the coast and higher over the SE Atlantic ocean (20° W–0, 10° S–5° N) than the climatological mean (Fig. 7). Dry convection over the Namibia-Kalahari dryland occurred during the end of deployment month. Moisture originated from the continent advected offshore by AEJ-S. The subsidence near the Namibian coast was enhanced in August 2017 by about 10~30 hPa day$^{-1}$ off the Namibian coast (~13° E, ~22° S) compared to the

climatological mean.

- *September 2016:* The AEJ-S was strong around 10S, ~ 4 km (i.e. 600– 650 hPa) and the correlation between AEJ-S and heat low was highest in September 2016. The continental ascent was strongest over the south of 15° S and the maximum AEJ-S occurred around 10° S. The zonal and meridional extent of recirculation is also tied to the

intensity and extent of AEJ-S. The meridional extent of the recirculated (returning) winds was wider (~10°) than August (top panel of Fig. 25). The St. Helena high was widely extended over the South Atlantic, extending to the Benguela coast (mid panel of Fig. 25b). The AEJ-S was strongly developed along with the LLJ especially with a time lag over the SE Atlantic near the coast, and their association was strong in the south of SE Atlantic. Moist



plume with high RH originated from the land was associated with moist convection in the north (5–10° S) and dry convection in the south (15-25° S), and they are advected by AEJ-S. The highest positive correlation between low-CF and high LST was obtained. The strong low-CF tended to be associated with low BLH.

The intensity and structure of the AEJ-S in September 2016 was slightly weaker than the climatological mean. The LLJ was stronger in September 2016 (~1 m s$^{-1}$) compared to the climatological mean. In general, the BLH over the South Atlantic Ocean in September 2016 was higher (~ 100 m) than the climatological mean (Fig. 7). Dry convection over the Namibia-Kalahari dryland was very pronounced during this month of deployment. The subsidence near the Namibian coast and South Atlantic in September 2016 was extensively stronger than the climatological mean.

- ○ ***October 2018***: The AEJ-S was also strong around 10° S, ~ 4 km (i.e. 600– 650 hPa) and the correlation between AEJ-S and heat low was high before the moist convection develop over the land at the end of the deployment month. Increasing convection and moist air with high RH moved further south (south of 10° S) as season changes. The updraft over the continent was slightly weak, but the horizontal temperature gradient still maintained the AEJ-S and vertical motion associated with the jet. The extent and intensity of the mid-level anticyclone, characterized by recirculating flow (wind vector, top panel in Fig. 25) of the AEJ-S, was the widest. Moist plumes with high RH marched down and transported westward along with the AEJ-S. The low-CF was largest among other months (August and September). The high low-CF was weakly associated with the high LST and low BLH, although they are not mostly statistically significant. The AEJ-S–LLJ and low-CF–LLJ relationship are highest in October 2018, which is regarded due to a rapidly developing mid-latitude frontal system over the Southeastern Atlantic Ocean.

  The AEJ-S was slightly weaker than the climatological mean, but the difference was small (less than ±1 m s$^{-1}$). The LLJ was weaker in October 2018 (-2~ -3 m s$^{-1}$) compared to the climatological mean. The BLH over the South Atlantic Ocean in October 2018 was lower (100–200 m) than the climatological mean (Fig. 7). Furthermore, the BLH was lowest in October 2018 compared to other deployment months. The subsidence near the Namibian coast was also reduced in October 2018 about 10~30 hPa day$^{-1}$ off the Namibian coast (~13° E, ~22° S) compared to the climatological mean.

This paper provides meteorological context for interpreting the airborne aerosol measurements. The variability of the meteorological fields during the deployment is highly modulated by daily to weekly variability of large-scale circulation, subsidence, and small-scale to the synoptic-scale of the convective system. If "fast" moving dynamic disturbance such as mid-latitude wave intrusion or frontal system development is present, the relation between stratocumulus and "slower" processes such as large-scale subsidence, LTS, BLH, and LLJ will not be simple. While we examined several large-scale meteorological factors tied to aerosol transport and stratocumulus decks, the detailed investigation of the processes controlling stratocumulus decks, aerosol lifting and transport is beyond the scope of this study. Future work is still needed





for  focusing on the specific science questions such as: 1) the roles of anticyclones over South Atlantic and the recirculating mid-level anticyclone associated with AEJ-S on aerosol transport, 2) the impact of the mesoscale convective system (MCS)

on AEJ-S and the role of the biomass burning (BB) aerosols in modulating them, 3) the separation of the meteorological and aerosol induced impact on low-cloud, 4) the impact of SST on low-cloud, aerosols, and their interactions at different time scales, 5) the causal relationship between low cloud and other meteorological factors over the different ocean basins.

For example, local topography in the different ocean basins can lead to different characteristics of the LLJ. A simple comparison of the Benguela LLJ (by the Angolan-Namibian coastline in the SE Atlantic) with Chilean LLJ (by the north-

central coastline of Chile in the SE Pacific) during ORACLES deployment periods shows: 1) there are semi-permanent large-scale anticyclones associated with high SLP (Muñoz and Garreaud, 2005; Garreaud and Muñoz, 2005). 2) Benguela LLJ is stronger than Chilean LLJ, largely due to larger pressure gradient; Benguela LLJ pulls away from the coast leading to lower BLH in SE Atlantic, while Chilean LLJ follows along the coastline of Chile leading to higher BLH in SE Pacific, largely associated with coastal topography (Zuidema et al., 2009). 3) For both basins, the LLJ–low-CF relationship is weak

and statistically insignificant except for October 2018 over SE Atlantic. Interestingly here, the climatological LLJ over the SE Atlantic tend to increase from August to October, but the LLJ in August 2017 and September 2016 were anomalously stronger than October 2018. Therefore, the further studies regarding interactions of the LLJ with other meteorological factors, aerosols, and low cloud will be of interest.

Recent studies suggest that stratocumulus clouds are crucial for assessing uncertainty in aerosol forcing (Bellouin et al.,

2020) and cloud feedbacks (Bretherton, 2015). This emphasizes the importance of advancing our knowledge of the role of aerosol, its transport, and its interaction with stratocumulus clouds in a changing climate.




**Data availability**

The analysis is based on the open source data and ORACLES P-3 flight track data. ERA 5 is open-source reanalysis data (https://www.ecmwf.int/en/forecasts/datasets/reanalysis-datasets/era5). MODIS Aqua (Terra) Level 3 product is from https://ladsweb.modaps.eosdis.nasa.gov/missions-and-measurements/products/MYD08_M3

(https://ladsweb.modaps.eosdis.nasa.gov/missions-and-measurements/products/MOD08_M3), and VIIRS Level3 cloud data is from (https://ladsweb.modaps.eosdis.nasa.gov/missions-and-measurements/products/CLDPROP_D3_VIIRS_SNPP). OISST data is from (http://www.remss.com/measurements/sea-surface-temperature/oisst-description/). TRMM (TRMM_3B43) monthly rainfall estimate data is also obtained through (https://disc.sci.gsfc.nasa.gov/datasets/TRMM_3B43_7/summary?keywords=TRMM). The ORACLES P-3 flight track data

can be obtained through ORALCES Science Team (2000a-c) references: https://doi.org/10.5067/Suborbital/ORACLES/P3/2018_V2 (ORACLES Science Team, 2020a), https://doi.org/10.5067/Suborbital/ORACLES/P3/2017_V2 (ORACLES Science Team, 2020b), https://doi.org/10.5067/Suborbital/ORACLES/P3/2016_V2 (ORACLES Science Team, 2020c).

**Author contributions**

RW and PZ envisioned the original ORACLES meteorological overview concept, and RU, JMR, and LP designed the manuscript structure. IC provided the climatological mean and monthly mean MODIS low-cloud data and assisted JMR to obtain the VIIRS daily mean cloud data product. LP and JMR developed the methodology of determining the BLH. JMR processed the data analysis and visualized the results. JMR, LP and RU interpreted results and JMR wrote the manuscript. LP, RU, PZ, RW, IC, and JR edited the manuscript. JR, RW, and PZ made critical contributions to the ORALCES field

campaigns. LP and RU led the meteorological forecast briefing during the whole ORALCES field campaigns.

**Competing interests**

Paquita Zuidema is a guest editor for the ACP Special Issue: "ACP special issue: New observations and related modelling studies of the aerosol–cloud–climate system in the Southeast Atlantic and southern Africa regions". The rest of the authors declare that they have no conflict of interest.

**Special issue statement**

This article is part of the special issue "New observations and related modelling studies of the aerosol-cloud-climate system in the Southeast Atlantic and southern Africa regions (ACP/AMT inter-journal SI)". It is not associated with a conference.

**Acknowledgements**





The authors give sincere gratitude to all the ORACLES participants who make the project successful. The first author gives
thanks to Sara Purdue for her assistance on obtaining OISST data, and Yohei Shinozuka for helpful discussion. The first
author is also thankful to Kristina Pistone and Samuel E. LeBlanc for providing the list of 3 years of ORACLES flight days
and targeted goals. ORACLES campaign was funded by NASA Earth Venture Suborbital-2 (NNH13ZDA001N-EVS2).



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
