# Peer review of "A meteorological overview of the ORACLES (ObseRvations of Aerosols above CLouds and their intEractionS) campaign over the southeast Atlantic during 2016-2018: Part 1 - Climatology"

_Atmospheric Chemistry and Physics, 2021_

## Author Comment (AC1)

**Response to reviewer 1**

In this response letter, we repeat the reviewer's questions and answer them one by one. Our response is marked in blue letters.

Review of the paper "A meteorological overview of the ORACLES (ObseRvations of Aerosols above CLouds and their intEractionS) campaign over the southeast Atlantic during 2016-2018" by Ryoo et al. 2021, submitted to Atmospheric Chemistry and Physics Discussions.

**General Comments**

The paper describes the atmospheric conditions during three field campaigns over southeast Atlantic in 2016-2018. The manuscript is well written and clear, figures quality is good and captions are informative. The authors provide a comprehensive picture of the climatological conditions, seasonal anomalies and synoptic evolution during the deployments, and physical mechanisms are correctly described and widely detailed. The paper appears to be specifically addressed to the community studying the physics of the atmosphere in the southeast Atlantic and provides a valuable contribution to all the teams involved in field campaigns during 2016-2018 for putting their observations into a synoptic context.

However, in my opinion the paper remains almost exclusively descriptive and does not respond to any relevant scientific question. In the main text, several interesting scientific questions are highlighted, but the authors do not investigate any of them, only speculating on possible (always plausible) explanations and requiring further investigation in future papers. For instance, I found very interesting the anomaly (almost disappearance and/or lifting) of the AEJ-S in August 2017, which would be worth to be investigated, in terms of both local and remote drivers.

This is my main and only concern. Despite the overall good quality of the paper itself, I am not sure it fits with the scope of ACP and/or the special issue. I list below a few general and specific suggestions on what I believe could be improved.

=> Thank you for your valuable comments. This is a meteorological overview paper, so our main focus was to describe the overall meteorological conditions during the 2016-2018 ORACLES campaigns. While investigating underlying mechanism with more depth was important, this was not our focus in this overview paper.

Following your suggestions, we trimmed lots of text and kept the key figures. Furthermore, since 1) both reviewers pointed out the length of the manuscripts and 2) the other reviewer suggested that we separate the paper into two parts (*part 1*: climatology at monthly scales; *part 2*: deployment month at daily to synoptic scales), we decided to separate the paper into two parts. In this revision, we will focus on the part 1 (climatology). We also examined further the possible environmental factors to cause the anomalous AEJ-S in August 2017 in this revision. We found a few distinctive characteristics during August 2017, but among them, we think upper-level disturbances play a large role in modulating the large-scale environment and thereby the AEJ-S. In order words, the position of the upper-level disturbance induces an anomalous circulation, thereby changing the cross-latitude temperature gradient over the land, ultimately reducing AEJ-S. We will describe this more in detail with figures in the response letter below as well as in the revised manuscript.

We have also further investigated a possible remote driver for weakening AEJ-S. Although beyond the scope of this paper to investigate further, previous other works indicates that wave activity supporting convection near the equator is weaker when the MJO is weak (Guo et al., 2014), and in turn the northern AEJ (AEJ-N), located to the north of the Congo, is also weakened (Ventrice and Thorncroft, 2013). Preliminary analysis showed that MJO convection was weak in August 2017 over Africa region. The impact of the MJO to the AEJ-S remains underexplored, however, and we can only speculate that a weaker MJO might also help weaken the AEJ-S. Furthermore, the mechanism of the magnitude of MJO influence on Southern Africa is still poorly characterized (Zaitchik, 2017).

More importantly, as mentioned earlier, this is an "overview" paper for better interpreting ORACLES measurements, so we want to keep focusing on this part, rather than investigating the detailed processes. Thus, we will briefly discuss this in the text without very detailed figures and explanations. Instead, we will put some results related to this remote driver in the supplementary materials.

The paper is very long, with many (25!) multi-panel figures. Figures are often the repetition of similar analysis, it is not easy for the reader to stay focused on the narrative of the paper. The authors could try select non-key information in text and figures and move it to the Supplement.

=> Both reviewers pointed this out, and we worked on this part. We trimmed the texts and tried to keep the main key parts. We also separate the paper into two parts following the second reviewer's suggestion.

In the Summary and Discussion, the authors speculate on the effect of meteorological conditions on the aerosol transport but no evidence of the actual effect is provided (while the cloud-circulation relationship is well described in the paper). I believe this is a key aspect which could provide added value to the paper. ECMWF CAMS reanalysis could be used to frame the aerosol patterns at the regional scale.

=> Thank you for the comments. Following the reviewer's suggestion, we also looked at the ECMWF CAMS reanalysis data. Since we will focus on climatology in this paper, we observed the features from the monthly mean data.

The BC and CO have similar features as in August 2017; particularly the BC mixing ratio is "lower" in August 2017 than the climatological mean, especially off the coast and over the ocean, shown in Fig. R1. We think this is reasonable because a weaker AEJ-S leads to weaker advection of these burning trace constituents out over the SE Atlantic. BC mixing ratio in September 2016 are slightly higher than the climatological mean, especially at the jet entrance region over land following the ascent.

---

## Author Comment (AC2)

**Response to reviewer 2**

**In this response letter, we repeat the reviewer's questions and answer them one by one. Our response is marked in blue letters.**

This paper is important and presents original work on a feature that has not received much attention. I was originally going to rate this as "minor revision". However, I realized that the length and complexity is such that very few would read it. I am extremely interested in this topic, but would dread, as a researcher, having to go through this article. The word count must be close to 15,000, roughly twice that accepted by most journals. It indicates 26 figures, yet most have numerous panels. In most cases those panels are quite independent of each other, so that 40 figures is a more realistic number. I am indicating major revision because I feel it is imperative that the authors break this into two articles. That is actually not that difficult or time-consuming to do. It will guarantee that the work receives more attention.

Based on this comment, I have stopped the review around Figure 5.

=> Thank you for your valuable comments. We appreciated your comments. Following your suggestion, we now separate the paper into two parts. The part 1 of the revised manuscript will be focused on the climatology part.

I do hope the authors will consider what is suggested above, as it is important work and should eventually be published. Also, the authors might want to look at the extensive work on the fogs and stratus done by Cermac and colleagues and cite some of that literature.

=> Thank you for your suggestion. We will go over the Cermak et al. papers and will cite them in the revised manuscript where appropriate.

**INDIVIDUAL COMMENTS**

The abstract is too long and includes too many details that really belong in the text. This is particularly the case in describing the anomalous characteristics for the three months considered.

=>. We will reduce the abstract for the revised manuscript. Reviewer 1 however requests more information be included on the cloud and aerosol anomalies within the abstract, so we do expect to keep information pertinent to the anomalies in the abstract.

Good overview of the background literature. Variables considered and data sets to derive them are clearly described.

=> Thank you for your comments.

A major concern is their use of ERA-5 because its representation of the AEJ-S is questionable. The core tends to be over the ocean in the various diagrams, with little extension over the land in most cases. The only paper that focuses directly on the AEJ-S is Kuete et al. (2020) in Climate Dynamics (see also Jackson et al. 2009). Of the three reanalyses Kuete et al. examined, only ERA-Interim shows a pattern similar to that in this paper. The others show a core further land-ward and extensive development over the land. The greater development over land is consistent with the idea that the temperature gradient between the rain forest to the north and the dry season in the savanna to the south is the cause of the

jet. I would suggest that the authors start by showing the jet in at least two additional reanalyses, in order to recognize that there are differences. MERRA and JRA 55 would be good choices. They should also speculate on how the use of ERA-5, with the core over the ocean, would affect their results.

=> Thank you for pointing this out. To comply with the reviewer's comments, we investigated the location of AEJ-S in the other reanalysis data such as MERRA2, JRA 55, and NCEP/NCAR for climatology and deployment month, using monthly mean data. The climatology is based on 2000-2018 for all reanalyses data. They have different spatial resolutions: ERA-5 has 0.25 deg x 0.25 deg, MERRA 2 has 0.625 deg x 0.5 deg, JRA55, and NCEP/NCAR reanalysis has 2.5 deg x 2.5 deg resolution, respectively.

All three reanalyses, ERA5, MERRA2, and JRA55 data, show similar features in terms of location and strength of AEJ-S, with the weakest magnitude shown within the most coarsely resolved reanalysis, namely the JRA55 (see Fig. R1). Perhaps, for this reason, the AEJ-S in August 2017 is almost the same as the climatological mean in JRA55 (with the core value slightly weaker). However, the local enhancement of the upper-level wind is evident within all three reanalyses mentioned above.

One small difference in ERA5 and other reanalyses is that the core of AEJ-S is slightly displaced to the coast in the higher resolution data such as ERA5 and MERRA2 in September and October, as the reviewer mentioned. We investigated this a bit more and found that this might be related to the variability of the AEJ-S core altitude at the jet entrance region and exit region represented in the monthly mean data in ERA5. The preliminary analysis showed that the core of AEJ-S using the monthly mean data in ERA5 tends to occur around 3.5 km (slightly lower than 600 hPa) at the jet entrance region over land, while it gets up to about 4 km (about the same as 600 hPa) at the jet exit region over the ocean, especially in October.

We also checked the zonal wind isotach at 650 hPa level and found that the core is located over land, especially in the climatological mean. When we looked at the 6-hourly wind data at 600 hPa, AEJ-S is certainly originated from the land in ERA 5 (not shown). Therefore, while the actual position in the monthly mean data may need to be interpreted with caution in different resolutions of data and different reanalysis products, we think the position of the AEJ-S in ERA5, MERRA2, and JRA55 is still reasonable.

---

## Author Response (AR1)

**Ms. No.: ACP-2021-274**

**A meteorological overview of the ORACLES (ObseRvations of Aerosols above CLouds and their intEractionS) campaign over the southeast Atlantic during 2016-2018**

Ju-Mee Ryoo et al.

**Major changes in the revised manuscript:**

- Following the suggestion of Reviewer #2, we have separated the paper into two parts to deliver key messages more effectively: Part1 (climatology) and Part2 (individual deployments). We focus on the first part (Part 1-Climatology) for this revision. Accordingly, the title is also changed into "A meteorological overview of the ORACLES (ObseRvations of Aerosols above CLouds and their intEractionS) campaign over the southeast Atlantic during 2016-2018: Part 1 Climatology".
- **Following the suggestion of Reviewer #1**, we have also investigated the possible reasons for the weaker AEJ-S in August 2017 compared to the climatological mean. How the aerosol from ECMWF CAMS reanalysis behaves along with the meteorological variables during the deployment months has been discussed.

We have taken the reviewers' comments and performed additional analyses based on reviewers' comments. We hope we have addressed all the key comments and suggestions from the reviewers and incorporated them in the revised manuscript. Enclosed is a point-by-point response to each reviewer's comments.

**Response to reviewer #1**

In this response letter, we repeat the reviewer's questions and answer them one by one. Our response is marked in blue letters.

Review of the paper "A meteorological overview of the ORACLES (ObseRvations of Aerosols above CLouds and their intEractionS) campaign over the southeast Atlantic during 2016-2018" by Ryoo et al. 2021, submitted to Atmospheric Chemistry and Physics Discussions.

**General Comments**

The paper describes the atmospheric conditions during three field campaigns over southeast Atlantic in 2016-2018. The manuscript is well written and clear, figures quality is good and captions are informative. The authors provide a comprehensive picture of the climatological conditions, seasonal anomalies and synoptic evolution during the deployments, and physical mechanisms are correctly described and widely detailed. The paper appears to be specifically addressed to the community studying the physics of the atmosphere in the southeast Atlantic and provides a valuable contribution to all the teams involved in field campaigns during 2016-2018 for putting their observations into a synoptic context.

However, in my opinion the paper remains almost exclusively descriptive and does not respond to any relevant scientific question. In the main text, several interesting scientific questions are highlighted, but the authors do not investigate any of them, only speculating on possible (always plausible) explanations and requiring further investigation in future papers. For instance, I found very interesting the anomaly (almost disappearance and/or lifting) of the AEJ-S in August 2017, which would be worth to be investigated, in terms of both local and remote drivers.

This is my main and only concern. Despite the overall good quality of the paper itself, I am not sure it fits with the scope of ACP and/or the special issue. I list below a few general and specific suggestions on what I believe could be improved.

=> Thank you for your valuable comments.

Following your suggestions, we trimmed lots of text and kept the key figures. Furthermore, since 1) both reviewers pointed out the length of the manuscripts and 2) the other reviewer suggested that we separate the paper into two parts (*Part 1*: climatology at monthly time scales; *Part 2*: deployment month at daily to synoptic time scales), we decided to separate the paper into two parts. In this revision, we focus on Part 1 - Climatology.

This is a meteorological overview paper, so our main focus was to describe the overall meteorological conditions during the 2016-2018 ORACLES campaigns. While investigating the underlying mechanism with more depth was important, this was not our focus in this overview paper. However, we examined further the possible environmental factors to cause the anomalous AEJ-S in August 2017, following your suggestion. We found a few distinctive characteristics during August 2017, but among them, we think upper-level disturbances play a significant role in modulating the large-scale environment and thereby the AEJ-S. In order words, the position (phase) of the upper-level disturbance induces an anomalous circulation, thereby changing the cross-latitude temperature gradient over the land, ultimately

reducing AEJ-S. We described this more in detail with figures in the response letter below as well as in Part 1 of the revised manuscript.

We have also further investigated a possible remote driver for weakening AEJ-S. As a remote driver, the Madden Julian Oscillation convection (MJO; Madden and Julian, 1994; Wheeler and Hendon, 2004), an intraseasonal convective variability in the equatorial troposphere with a periodicity of about 30-90 days, may contribute to the weakening of AEJ-S in August 2017, because this is weakened over Africa during this period (shown in Fig. 10S in the supplementary material). The MJO can affect the timing and intensity of convectively coupled Kelvin waves and convective activity over Africa (Guo et al., 2014), which can affect AEJ-S activity (Ventrice and Thorncroft, 2013; Zaitchik, 2017). However, this remote driver has been investigated for the AEJ-N, and the MJO's influence on the AEJ-S is less understood and remains unclear.

The paper is very long, with many (25!) multi-panel figures. Figures are often the repetition of similar analysis, it is not easy for the reader to stay focused on the narrative of the paper. The authors could try select non-key information in text and figures and move it to the Supplement.

=> Both reviewers pointed this out, and we worked on this part. We trimmed the texts and tried to keep the main key messages. We also separated the paper into two parts following the second reviewer's suggestion.

In the Summary and Discussion, the authors speculate on the effect of meteorological conditions on the aerosol transport but no evidence of the actual effect is provided (while the cloud-circulation relationship is well described in the paper). I believe this is a key aspect which could provide added value to the paper. ECMWF CAMS reanalysis could be used to frame the aerosol patterns at the regional scale.

=> Thank you for the comments. Following the reviewer's suggestion, we also looked at the ECMWF CAMS reanalysis data. Since we focused on climatology in this paper, we observed the features from the monthly mean data.

The black carbon mixing ratios (BC) and Carbon Monoxide mixing ratios (CO) have similar features as in August 2017; particularly the BC is "lower" in August 2017 than the climatological mean, especially off the coast and over the ocean, shown in Fig. R1. We think this is reasonable because a weaker AEJ-S leads to weaker advection of these burning trace constituents out over the SE Atlantic. BC in September 2016 is slightly higher than the climatological mean, especially at the jet entrance region over land following the ascent (See Fig. R2).